# SPATIO-TEMPORAL FEW-SHOT LEARNING VIA DIFFUSIVE NEURAL NETWORK GENERATION

**Yuan Yuan**[*], **Chenyang Shao**[*], **Jingtao Ding**[†] **Depeng Jin, Yong Li**[†]
Department of Electronic Engineering,
BNRist, Tsinghua University,
Beijing, China
`y-yuan20@mails.tsinghua.edu.cn`, `{dingjingtao, liyong07}@tsinghua.edu.cn`

## ABSTRACT

Spatio-temporal modeling is foundational for smart city applications, yet it is often hindered by data scarcity in many cities and regions. To bridge this gap, we propose a novel generative pre-training framework, GPD, for spatio-temporal few-shot learning with urban knowledge transfer. Unlike conventional approaches that heavily rely on common feature extraction or intricate few-shot learning designs, our solution takes a novel approach by performing generative pre-training on a collection of neural network parameters optimized with data from source cities. We recast spatio-temporal few-shot learning as pre-training a generative diffusion model, which generates tailored neural networks guided by prompts, allowing for adaptability to diverse data distributions and city-specific characteristics. GPD employs a Transformer-based denoising diffusion model, which is model-agnostic to integrate with powerful spatio-temporal neural networks. By addressing challenges arising from data gaps and the complexity of generalizing knowledge across cities, our framework consistently outperforms state-of-the-art baselines on multiple real-world datasets for tasks such as traffic speed prediction and crowd flow prediction. The implementation of our approach is available: `https://github.com/tsinghua-fib-lab/GPD`.

## 1 INTRODUCTION

Spatio-temporal prediction is a fundamental problem in various smart city applications (Xia et al., 2024; Zhou et al., 2024; Wang et al., 2023a;c;b). Many deep learning models are proposed to solve this problem, whose successes however rely on large-scale spatio-temporal data. Due to imbalanced development levels and different data collection policies, urban spatio-temporal data, such as traffic and crowd flow data, are usually limited in many cities and regions. Under these circumstances, the model's transferability under data-scarce scenarios is of pressing importance.

To address this issue, various transfer learning approaches have emerged for spatio-temporal modeling. Their primary goal is to leverage knowledge and insights gained from one or multiple source cities and apply them effectively to a target city. These approaches can be broadly classified into two main categories. (1) Coarse-grained methods consider each city as a unified entity and transfer the learned knowledge at the city level (Pang et al., 2020; He et al., 2020; Ding et al., 2019; Tang et al., 2022). (2) Fine-grained methods dissect cities into smaller regions to enable a more refined exploration of region-level knowledge (Wang et al., 2019; Guo et al., 2018; Yao et al., 2019; Liu et al., 2021; Jin et al., 2022; Lu et al., 2022), which have shown better performance due to the inherent disparities between source and target cities. However, existing fine-grained methods largely rely on elaborated matching designs, such as utilizing auxiliary data for similarity calculation (Wang et al., 2019) or incorporating multi-task learning to obtain implicit representations (Lu et al., 2022). How to enable a more general knowledge transfer to automated retrieving similar characteristics across source and target cities still remains unsolved.

---

[*]Equal contribution.
[†]Corresponding author.

Recently, pre-trained models have yielded significant breakthroughs in the fields of Natural Language Processing (NLP) (Brown et al., 2020; Vaswani et al., 2017). Prompting techniques are also introduced to reduce the gap between fine-tuning and pre-training (Brown et al., 2020). At its core, the adoption of pre-trained models embodies the fundamental principles of transfer learning, allowing the model to acquire a broad understanding of various patterns and subsequently adapt to address specific tasks. Nowadays, what's particularly noteworthy is that advanced pre-trained models no longer require laborious fine-tuning, but leverage effective prompting techniques for fast adaptation (Brown et al., 2020; Rombach et al., 2022). Such capability serves as a crucial remedy for the current limitations in spatio—temporal few-shot learning, which offers the potential to enhance fine-grained transfer by general matching techniques.

However, despite these remarkable advancements of pre-trained models, there remains a notable gap in developing pre-trained models tailored for spatio-temporal scenarios. This disparity can be attributed to several challenges. Firstly, NLP benefits from a shared vocabulary that can be applied across various scenarios or tasks, while urban areas across different cities are geographically disjoint, lacking common elements that enable straightforward knowledge transfer. Secondly, substantial divergence often exists in data distributions between source and target cities, leading to the potential noise or even counterproductive information in the knowledge acquired from source cities (Jin et al., 2022). Additionally, pattern divergence exists even within one city due to regions with different functions. These divergences pose challenges to the effective transfer of knowledge to a target city. In other words, the knowledge obtained from the data often contains noise and bias, making it hard to train a universal model that perfectly fits patterns with high variations in different cities. Achieving effective transfer in this context requires addressing a more complex setting of generalization across cities. The key challenge, therefore, lies in determining what transferable knowledge, analogous to the shared semantic structure in NLP, can be established for urban settings.

In this work, we present a generative pre-training framework for spatio-temporal few-shot learning. Instead of fitting the spatio-temporal data with a unified model, we propose a novel pre-training strategy that captures universal patterns from optimized neural network parameters. In simpler terms, we recast spatio-temporal few-shot learning as pre-training a generative hypernetwork. This hypernetwork is designed to adaptively generate unique parameters for spatio-temporal prediction models guided by prompts.

Essentially, our pre-training approach empowers the capability to adaptively generate distinctive neural networks in response to diverse data distributions, which addresses the challenges arising from data gaps across cities or regions. To elaborate, we begin with a set of neural networks optimized for spatio-temporal predictions. Then we design a Transformer-based diffusion model to generate network parameters from Gaussian noise conditioned on the prompt. We also design a class of conditioning strategies for the prompt to guide the denoising process. Consequently, when presented with a target prompt encoding spatio-temporal characteristics of the target scenario, the diffusion model generates corresponding neural networks for accurate predictions. Our framework is model-agnostic, ensuring compatibility with state-of-the-art spatio-temporal prediction models. We summarize our contributions as follows:

- We propose to leverage pre-training paradigm to achieve effective fine-grained spatio-temporal knowledge transfer across different cities, which stands as a pioneering practice in handling urban data-scarce scenarios with pretrained models.

- We propose a novel **G**enerative **P**re-training framework based on **D**iffusion models, called GPD. It leverages a Transformer-based diffusion model and city-specific prompts to generate neural networks, opening new possibilities for improving spatio-temporal modeling.

- Extensive experiments on multiple real-world scenarios demonstrate that GPD achieves superior performance towards data-scarce scenarios with an average improvement of 7.87% over the best baseline on four datasets.

## 2 RELATED WORKS

**Spatio-Temporal Few-Shot Learning.** Addressing data scarcity is a pervasive challenge in machine learning-driven urban computing applications (Wei et al., 2016; He et al., 2020; Zhou et al., 2023a). This issue is particularly pronounced in cities lacking advanced digital infrastructure or in

the initial stages of deploying sensors, as they struggle to accumulate adequate data. To solve these issues, spatio-temporal few-shot learning (Zhuang et al., 2020) has emerged as a promising solution. Existing solutions can be broadly categorized into coarse-grained and fine-grained methods. Coarse-grained methods (Fang et al., 2022; Liu et al., 2018; Pang et al., 2020; He et al., 2020; Ding et al., 2019; Tang et al., 2022; Yao et al., 2019) treat each city as a whole and apply transfer learning techniques, such as adversarial learning (Fang et al., 2022) and meta-learning (Yao et al., 2019), to leverage knowledge across urban contexts. Differently, fine-grained methods (Wang et al., 2019; Guo et al., 2018; Liu et al., 2021; Jin et al., 2022) divide cities into smaller regions and identify similar region pairs for knowledge transfer. They use various techniques, such as Similarity-based matching (Wang et al., 2019) or re-weighting (Jin et al., 2022), to facilitate the transfer of insights between these regions. Fine-grained methods have demonstrated better performance due to the inherent disparities between source and target cities (Jin et al., 2022). (Lu et al., 2022) proposed to learn meta-knowledge to facilitate node-level knowledge transfer. Instead, we propose a generative pre-training framework made up of a diffusion-based hypernetwork and adaptive conditioning strategy. Our framework not only exhibits powerful parameter generation capabilities but also offers remarkable flexibility by allowing the use of various forms of prompts.

**Diffusion Models.** Diffusion probabilistic models (Ho et al., 2020; Song et al., 2020; Nichol & Dhariwal, 2021) have emerged as a powerful alternative to generation tasks, which not only outperform Generative Adversarial Networks (Goodfellow et al., 2020) on image generation tasks with higher fidelity (Dhariwal & Nichol, 2021; Rombach et al., 2022), but also enable effective cross-modal conditioning (Bao et al., 2023; Nichol et al., 2022). In addition to image generation, diffusion models have also been utilized for other tasks, such as video generation (Ho et al., 2022; Luo et al., 2023), text generation (Li et al., 2022; Gong et al., 2022), implicit neural fields generation (Erkoç et al., 2023), network learning (Peebles et al., 2022), graph generation (Vignac et al., 2023), and spatio-temporal prediction (Wen et al., 2023; Yuan et al., 2023). In contrast, our approach leverages diffusion models to generate neural network parameters of spatio-temporal prediction models.

**Hypernetworks.** Hypernetworks represent a class of generative models tasked with producing parameters for other neural networks (Ha et al., 2017). These hypernetworks serve two primary purposes: (1) the generation of task-specific parameters (Ha et al., 2017; Alaluf et al., 2022), where hypernetworks are usually jointly trained with specific task objectives, and (2) the exploration of neural network characteristics (Schürholt et al., 2022; 2021), which focus on learning representations of neural network weights to gain insights of network properties. In our work, we align with the first purpose, focusing on the generation of parameters tailored to the target city for spatio-temporal prediction. There are also hypernetworks for spatio-temporal prediction (Pan et al., 2019; Bai et al., 2020; Pan et al., 2020; 2018; Li et al., 2023), which use non-shared parameters for different nodes. Our key differences lie in three aspects. Firstly, our framework ensures compatibility with state-of-the-art models, while those works propose specific model designs. Secondly, different from the pre-training solution (Li et al., 2023) with a mask autoencoder, we focus on the pre-training of model parameters. Thirdly, our approach is a spatio-temporal few-shot learning framework, capable of acquiring knowledge from multiple cities and effectively transferring it to few-shot scenarios. In contrast, those works have certain limitations in data-scarce urban environments.

# 3 PROPOSED METHOD

## 3.1 PRELIMINARY

We introduce some preliminaries related to our research problem.

**Definition 1** (Spatio-Temporal Graph). *A spatio-temporal graph (STG) as $\mathcal{G}_{ST} = (\mathcal{V}, \mathcal{E}, \mathcal{A}, \mathcal{X})$, where $\mathcal{V}$ is the node set, $\mathcal{E}$ is the edge set, $\mathcal{A}$ is the adjacency matrix, and $\mathcal{X}$ denotes the node feature. The node features can encompass various aspects such as crowd flows and traffic speed.*

For example, for the crowd flow dataset, nodes correspond to segmented regions within the city, with edges indicating their geographic adjacency. For the traffic speed dataset, nodes represent the sensors deployed along the road network, and edges signify the connections between these sensors. The construction of the spatio-temporal graph is adaptable to the specific requirements of each task.

**Problem 1** (Spatio-Temporal Prediction). *For a spatio-temporal graph $\mathcal{G}_{ST}$, suppose we have $L$ historical signals $[X^{t-L+1}, X^{t-L+2}, \ldots, X^t]$ for each node and predict the future $n$ steps*

**(a) Preparation of Neural Networks**   **(b) Pre-Train the Diffusion Model**   **(c) Generate Target Parameters**

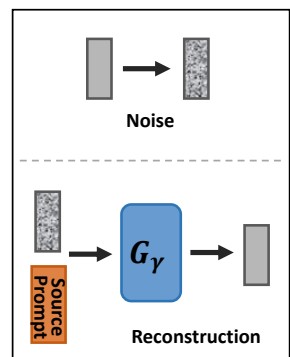
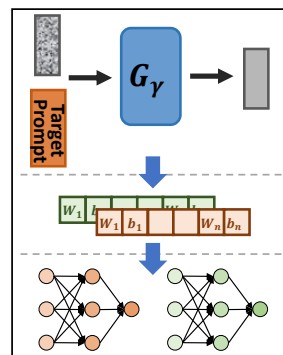

Figure 1: An overview of the proposed framework. (a) A collection of optimized spatio-temporal prediction models based on the dataset of source cities; each model's parameters are transformed into a vector-based format. (b) Pre-training the diffusion model to generate neural network parameters from the noise given the prompt. (c) Utilizing the pre-trained diffusion model to generate neural network parameters for the target city based on the target prompt.

$[X^{t+1}, \ldots, X^{t+n}]$. *The prediction task is formulated as learning a $\theta$-parameterized model $F$ given a spatio-temporal graph $\mathcal{G}_{ST}$:* $[X^{t+1}, \ldots, X^{t+n}] = F_\theta(X^{t-L+1}, X^{t-L+2}, \ldots, X^t; \mathcal{G}_{ST})$.

**Problem 2** (Spatio-Temporal Few-Shot Learning). *Spatio-temporal few-shot learning is formulated as learning knowledge $\mathcal{K}$ from source cities $\mathcal{C}_{1:P}^{source} = \{\mathcal{C}_1^{source}, \ldots, \mathcal{C}_P^{source}\}$, and then transfer the learned knowledge $\mathcal{K}$ to the target city with few-shot structured data to facilitate spatio-temporal predictions.*

**Problem 3.** *We formulate the spatio-temporal few-shot learning as pre-training a diffusion model to conditionally generates neural network parameters, and then utilizing it to generate parameters for the target city's prediction model. Suppose we have a collective of learned prediction models $F = \{F_{\theta_1}, F_{\theta_2}, \ldots, F_{\theta_N}\}$ from a set of data-rich source cities $\mathcal{C}_{1:P}^{source} = \{\mathcal{C}_1^{source}, \ldots, \mathcal{C}_P^{source}\}$, we aim to pre-train a diffusion model to generate $F$ with source-city prompts. The optimized diffusion model serves as the learned knowledge $\mathcal{K}$, which can be transferred to the target city.*

### 3.2 OVERALL FRAMEWORK

Figure 1 provides an illustrative overview of our proposed framework, which contains three phases. The left part shows the preparation of model parameters, from which we can obtain a collection of optimized neural network parameters. The middle part illustrates the pre-training phase within source cities, where the diffusion model $G_\gamma$ is trained to generate meaningful parameters from Gaussian noise given source prompts. The right part demonstrates the how we transfer the learned diffusion model to facilitate spatio-temporal predictions in the target city. In the following subsections, we elaborate on the details of these phases.

### 3.3 SPATIO-TEMPORAL PREDICTION MODEL

Our framework is model-agnostic, ensuring compatibility with state-of-the-art spatio-temporal prediction models. Here we utilize well-established STG models, specifically STGCN (Yu et al., 2017) and GWN (Wu et al., 2019), and MLP-based model STID (Shao et al., 2022a) , as prediction models[1]. In this work, we train the prediction model (denoted as $F_{\theta_i}$) separately for each region within a given city.

### 3.4 GENERATIVE PRE-TRAINING ON PARAMETER SPACE

Our approach is a conditional generative framework designed to directly learn from model parameters of source cities. This pre-training process enables the generation of new model parameters for target cities. Our training paradigm encompasses a three-phase approach as follows.

---

[1]Appendix A.4 provides more details of the three models.

**Preparation of Neural Networks.** As the initial step, we optimize spatio-temporal prediction models for each region within source cities and save their optimized network parameters. It's important to note that, while we use the same network architecture for different regions, each set of model parameters is uniquely optimized for its respective region. In other words, there is no parameter sharing, and each model is meticulously trained to achieve peak performance specific to its region. This optimization process can be formulated as follows:

$$\theta_i = \arg\min_{\theta_i} \sum_t \left\| f_{\theta_i}(X_i^{t-L+1}, X_i^{t-L+2}, \ldots, X_i^t; \mathcal{G}_{ST}) - [X^{t+1}, \ldots, X^{t+n}] \right\|^2,$$

$$\mathcal{T}(\theta) = \{\theta_1, \theta_2, \ldots, \theta_M\}, M = \sum_{s=1,\ldots,N} n_s,$$

where $i$ denotes the $i_{th}$ region within a source city, and $\theta_i$ represents parameters of its dedicated prediction model. $N$ denotes the total number of source cities, and $n_s$ indicates the number of regions in source city $s$. $\mathcal{T}(\theta)$ encompasses the optimized parameters of prediction models.

To optimize these parameters, we employ the standard Adam optimizer. These meticulously optimized parameters serve as the ground truth for the subsequent generative pre-training process. Algorithm 1 in Appendix A.5 illustrates the training process. Then we transform the parameters of each model into a vector-based format.

**Generative Pre-Training of the Diffusion Model.** Using the dataset comprising pre-trained parameters of prediction models and region prompts, we employ a generative model denoted as $G_\gamma$ to learn the process of generating model parameters in a single pass. Specifically, $G_\gamma$ predicts the distribution of parameters $p_G(\theta_i|p_i)$, where $p_i$ corresponds to the prompt of region $i$. We adopt diffusion models (Ho et al., 2020) as our generative model due to its efficacy in various generation tasks (Ho et al., 2022; Li et al., 2022; Vignac et al., 2023). Moreover, it has shown superior performance on multi-modal conditional generation (Bao et al., 2023; Nichol et al., 2022; Saharia et al., 2022).

We train the diffusion model to sample parameters by gradually denoising the vector from the Gaussian noise. This process is intuitively reasonable as it intriguingly mirrors the optimization journey from random initialization which is a well-established practice in existing optimizers like Adam. Our model takes two parts as the input: a noise-corrupted parameter vector $\theta^k$ and a region prompt $p$, with $k$ representing the step in the forward diffusion process. The training objective is as follows:

$$\mathcal{L} = \mathbb{E}_{\theta^0, \epsilon \sim \mathcal{N}(0,1), k}[\|\epsilon - \epsilon_\gamma(\boldsymbol{\theta}^k, p, k)\|^2], \tag{1}$$

where $\epsilon$ denotes the noise to obtain $\theta^k$ from $\theta^0$. Algorithm 2 illustrates the pre-training procedure.

**Sampling.** After pre-training, we can generate parameters $\theta_i$ by querying $G_\gamma$ using a region prompt $p$ for the target cities. When the region prompt in the target city closely resembles a region in multiple source cities, we can seamlessly approximate the model parameters specific to the target domain. In essence, we harness the power of the prompt to facilitate efficient knowledge transfer and precise parameter matching, which leverages the inherent similarities between regions across cities. The generation is an iterative sampling process from step $k = K$ to $k = 0$, which denoises the Gaussian noise into meaningful parameters. The generation process is formulated as follows:

$$\boldsymbol{\theta}^{k-1} = \frac{1}{\sqrt{\alpha_k}}(\boldsymbol{\theta}^k - \frac{\beta_k}{\sqrt{1 - \overline{\alpha}_k}}\epsilon_\gamma(\boldsymbol{\theta}^k, p, k)) + \sqrt{\sigma_\gamma}\mathbf{z}, \tag{2}$$

where $\mathbf{z} \sim \mathcal{N}(\mathbf{0}, \boldsymbol{I})$ for $k > 1$ and $\mathbf{z} = \mathbf{0}$ for $k = 1$. Algorithm 3 illustrates the procedure.

## 3.5 ARCHITECTURE

The generative model is a transformer-based diffusion architecture that learns the relationships across different layers of the prediction model via effective self-attention. It has been shown that the Transformer can effectively capture relationships of each token in long sequences. We find it to be a good choice to learn inter-layer and intra-layer interactions.

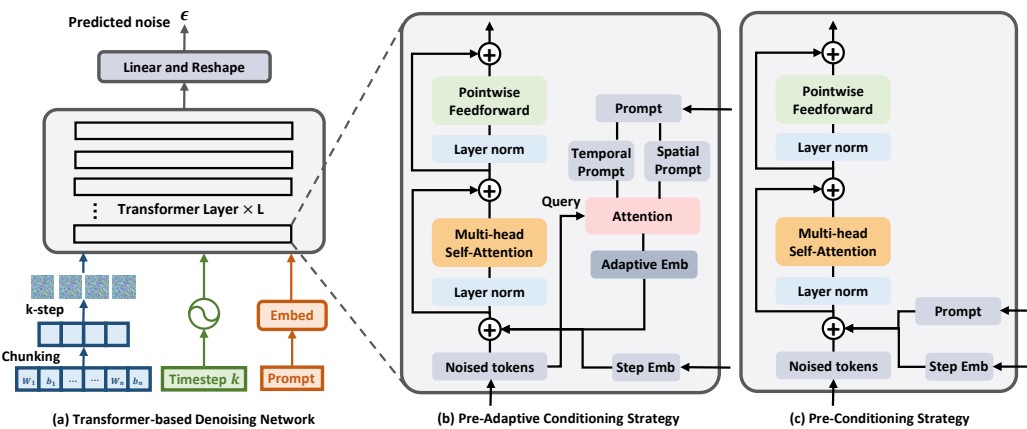

Figure 2: The network architecture of the denoising network. (b) and (c) illustrate two conditioning strategies, and we provide other used conditioning strategists in Appendix A.3.

**Parameter Tokenizers.** Spatio-temporal prediction models always exhibit complicated architectures consisting of various neural network layers, such as temporal layers, spatial layers, and spatio-temporal interaction layers. Usually, these layers are characterized by different shapes. These heterogeneous layer structures pose a challenge if we want to leverage the Transformer model to capture intricate relationships between them. To provide a clearer conceptual understanding, Table 4 and Table 5 in Appendix A.4 illustrate the layer structures of an spatio-temporal model. It is evident that different layers possess parameter tensors of diverse shapes. Therefore, it is necessary to decompose these parameters into vector-based tokens of uniform dimensions while preserving the inherent connectivity relationships within the original prediction model.

We achieve this by determining the greatest common divisor (denoted as $g$) of the parameters amount across all layers, then we perform layer segmentation by reshaping each layer into several tokens as: $n_m = N_m/g$, where $n_m$ is the number of tokens for the layer $m$ and $N_m$ is the number of parameters of the layer $m$. After that, the tokens are connected in a sequential manner, ensuring that layers that are adjacent in the network structure also maintain adjacency within the resulting sequence. This systematic approach facilitates the effective utilization of Transformer-based architectures in handling spatio-temporal prediction models with heterogeneous layer structures.

**Region Prompt.** The selection of region prompts offers flexibility, as long as they can capture the distinctive characteristics of a specific region. Various static features, such as population, region area, function, and distribution of points of interest (POI), can be leveraged for this purpose. In this work, we utilize region prompts from two aspects: spatial and temporal domains. For the spatial prompt, we utilize node embeddings extracted from a pre-trained knowledge graph. It only utilizes relations like region adjacency and functional similarity, which are easily obtained in all cities. Simultaneously, we employ self-supervised learning techniques (Shao et al., 2022b) on the very limited time series data (three days) to derive the embedding for the temporal prompt. Appendix A.2 provides more details of the prompt design.

**Denoising Network.** Figure 2 demonstrates the architecture of the denoising network, which adopts a prompt-based transformer diffusion model. After the layer segmentation, the parameters are restructured into a token sequence. In the denoising process, the transformer layers operate on the noised token sequence. In addition to the noised sequence, our transformer diffusion model also takes in the timestep $k$ and region prompt $p$. We anticipate that the model is capable of generating outputs conditioned on the region prompt. To this end, we explore several conditioning approaches, which introduce small but important modifications to the standard transformer layer design. The conditioning strategies are shown in Figure 2(b) and 2(c). We also explore more conditioning strategies, such as "Post-adaptive conditioning" and "Adaptive norm conditioning" (see Appendix A.3).

- **Pre-conditioning.** "Pre" denotes that the prompt is integrated into the token sequence before being fed into self-attention layers. In this conditioning strategy, we simply add the spatial prompt and temporal prompt into a spatio-temporal prompt. Then, we uniformly add it to each token. This design allows us to leverage standard transformer layers without requiring any modifications.

- **Pre-conditioning with inductive bias.** When adding the vector embeddings of $k$ and $p$ to the token embeddings, we introduce inductive bias: the timestep vector is added uniformly to each token, while the spatial prompt and temporal prompt are incorporated into spatial-related and temporal-related parameters, respectively. Inductive bias can be introduced flexibly based on the used spatio-temporal model architectures.

- **Pre-adaptive conditioning.** In this variant, the operation on the timestep embedding remains the same. However, concerning the prompt, we treat the embeddings of $p$ as a two-element sequence. An attention layer is introduced to realize the "adaptive" mechanism, dynamically determining to what extent the prompt should be added to specific token embeddings. This approach aims to empower the model to learn how to adaptively utilize the prompts, enhancing its conditioning capabilities.

## 4 EXPERIMENTS

### 4.1 EXPERIMENTAL SETUP

**Datasets.** We conduct experiments on two types of spatio-temporal prediction tasks: crowd flow prediction and traffic speed prediction. As for crowd flow prediction, we conducted experiments on three real-world datasets, including New York City, Washington, D.C., and Baltimore. Each dataset contains hourly urban flow of all regions, and we use half of the day (12 hours) as a training sample. As for traffic speed prediction, we conduct experiments on four real-world datasets, including Meta-LA, PEMS-BAy, Didi-Chengdu, and Didi-Shenzhen. Their time intervals are 5min, 5min, 10min, and 10min. We use 12 points as a training sample. For both tasks, we categorized the datasets into source cities and one target city. For example, if one specific city is set as the target dataset, we assume access to only a limited quantity of data, such as three-day data (existing models usually require several months of data to train the model). The diffusion model is trained using the other source cities with rich data. This same division and training strategy is consistently applied when targeting different cities. Appendix B.1 provides more details of the used datasets.

**Baselines and Metrics.** To evaluate the performance of our proposed framework, we compare it against classic models and state-of-the-art urban transfer approaches, including History Average (HA), ARIMA, RegionTrans (Wang et al., 2019), DASTNet (Tang et al., 2022), AdaRNN (Du et al., 2021), MAML (Finn et al., 2017), TPB (Liu et al., 2023), and ST-GFSL (Lu et al., 2022). We use two widely-used regression metrics: Mean Absolute Error (MAE) and Root Mean Squared Error (RMSE). Appendix B.2 and B.3 provide more details of baselines and implementation.

### 4.2 RESULTS

**Performance Comparison.** We compare the performance of our proposed GPD with baselines on two tasks: crowd flow prediction and traffic speed prediction. In both tasks, our goal is to transfer knowledge from the source cities to the target city. Due to space limits, Table 1 only reports the comparison results for the mean error of prediction over the subsequent 6 steps for some datasets. Comprehensive evaluations, including step-wise performance on all datasets, can be found in Appendix C. Based on these results, we have these noteworthy observations:

- **Consistent Superiority.** GPD consistently achieves the best performance compared with baseline approaches. When compared against the best-performing baselines (indicated by underlining in Table 1), GPD achieves an average error reduction of 4.31%, 17.1%, 2.1% and 8.17% in terms of MAE for Washington D.C., Baltimore, LA, and Chengdu. These improvements indicate that GPD enables effective knowledge transfer regarding parameter generation.

- **Competitive Long-Term Prediction.** GPD exhibits its competitiveness particularly in long-term prediction scenarios, as shown in Tables in Appendix C. For example, when considering Baltimore as the target city (Appendix Table 11), compared with the state-of-the-art baseline STGFSL, GPD shows an improvement in MAE performance of up to 22.1% at the $6_{th}$ step, compared to a 5.9% improvement at the $1_{st}$ step. This notable trend can be attributed to the inherent design of our framework, which facilitates the transfer of long-term temporal knowledge to the target city.

**Performance across Cities.** To investigate the influence of different source cities on our experimental outcomes, we conducted a series of experiments using various source cities for the target

| Model | Washington D.C. | | Baltimore | | LA | | Didi-Chengdu | |
|---|---|---|---|---|---|---|---|---|
| | MAE | RMSE | MAE | RMSE | MAE | RMSE | MAE | RMSE |
| HA | 21.520 | 47.122 | 15.082 | 26.768 | 3.257 | 6.547 | 3.142 | 4.535 |
| ARIMA | 19.164 | 42.480 | 12.902 | 23.758 | 7.184 | 10.91 | 5.381 | 7.087 |
| RegionTrans | 13.853 | 31.951 | 7.375 | 13.715 | 3.349 | 5.728 | 2.828 | 4.130 |
| DASTNet | 14.808 | 32.219 | 7.532 | 13.975 | 3.236 | 5.603 | 2.810 | 3.953 |
| MAML | 13.930 | 31.009 | 7.978 | 14.901 | 3.257 | 5.558 | 2.874 | 4.041 |
| TPB | 13.552 | 31.024 | 7.421 | 14.011 | 3.198 | 5.574 | 2.654 | 3.709 |
| STGFSL | 12.224 | 28.625 | 7.365 | 13.746 | 3.184 | 5.577 | 2.570 | 3.832 |
| GPD (ours) | **11.697** | **27.308** | **6.110** | **11.669** | **3.119** | **5.450** | **2.366** | **3.393** |
| Reduction% | **-4.31%** | **-4.62%** | **-17.1%** | **-14.9%** | **-2.1%** | **-3.2%** | **-8.17%** | **-8.62%** |

Table 1: Performance comparison of few-shot scenarios on two crowd flow datasets (Washington D.C. and Baltimore) and two traffic speed datasets (LA and Didi-Chengdu) in terms of MAE and RMSE. We use the average prediction errors over six steps as the result. Bold denotes the best results and underline denotes the second-best results.

| Target city | Source city | MAE | | | | | |
|---|---|---|---|---|---|---|---|
| | | Step 1 | Step 2 | Step 3 | Step 4 | Step 5 | Step 6 |
| Washington D.C. | B | 29.039 | 28.226 | 29.358 | 28.146 | 30.026 | 31.268 |
| | N | 27.569 | 28.631 | 27.285 | 28.229 | 27.858 | 29.667 |
| | B+N | 26.679 | 26.624 | 26.27 | 26.262 | 26.639 | 26.112 |
| Baltimore | W | 15.492 | 15.676 | 15.575 | 17.213 | 17.915 | 17.338 |
| | N | 17.234 | 17.737 | 17.395 | 17.325 | 18.734 | 18.179 |
| | W+N | 15.259 | 14.219 | 14.309 | 15.334 | 14.68 | 14.876 |
| NYC | W | 51.344 | 54.478 | 50.078 | 49.87 | 48.612 | 52.143 |
| | B | 53.145 | 51.088 | 48.292 | 54.658 | 62.494 | 60.525 |
| | W+B | 42.337 | 42.392 | 42.417 | 42.304 | 42.312 | 42.28 |

Table 2: Performace comparison with different source cities for crowd flow predictions. "W", "B", and "N" represent Washington D.C., Baltimore, and NYC, respectively.

city. For instance, when considering Washington D.C. as the target city, we analyzed three distinct source city scenarios: (1) using New York City (N) as the source city, (2) utilizing Baltimore (B) as the source city, and (3) incorporating both NYC and Baltimore (N+B) as source cities. The results summarized in Table 2 indicate that incorporating data from multiple source cities offers substantial benefits for both short-term and long-term predictions across all three target cities. This suggests that our framework effectively learned useful and transferable knowledge from these source cities. In one-step predictions, the inclusion of two source cities, as opposed to just one, results in performance improvements: 8.1% and 3.2% for Washington, 1.5% and 11.4% for Baltimore, and 17.5% and 20.3% for NYC. When extending our analysis to six-step predictions, the advantages of utilizing two source cities over one are even more pronounced, with performance improvements of 16.4% and 12.0% for Washington, 14.2% and 18.1% for Baltimore, and 18.9% and 30.1% for NYC. These results emphasize the significance of leveraging data from multiple source cities for the pre-training.

## 4.3 FLEXIBILITY OF GPD

Our framework is designed to be model-agnostic, providing support for cutting-edge spatio-temporal models. To validate the versatility of our framework, we also integrated another two advanced spatio-temporal prediction models, Graph WaveNet (GWN) (Wu et al., 2019) and Spatial-Temporal Identity (STID) (Shao et al., 2022a), into our framework. Specifically, STGCN and GWN are graph-based architectures, while STID does not leverage graph neural networks.

Figure 7 in Appendix presents a comprehensive performance comparison across the three spatio-temporal models—STGCN, GWN, and STID—across four datasets encompassing both crowd flow prediction and traffic speed prediction. The results reveal that STGCN and GWN exhibit nearly comparable performance, with GWN slightly outperforming. STID, given its relatively simpler design

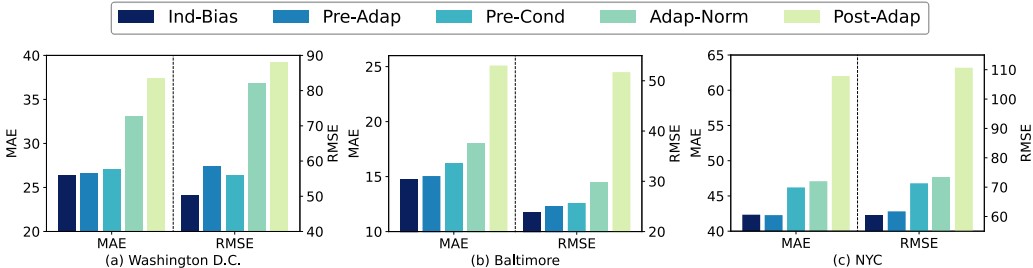

Figure 3: Performance comparison across different conditioning strategies.

and potential limitations in modeling complex spatio-temporal relationships, lags slightly behind STGCN and GWN. Therefore, we implement GWN as the base model for baselines to examine the effectiveness of our framework. Appendix C.1 provides the detailed comparison results. This comparison underscores our framework's adaptability to integrate existing spatio-temporal models and its inherent potential to enhance prediction performance, particularly as more powerful models become available.

## 4.4 IN-DEPTH STUDY OF GPD

**Conditioning Strategy.** We investigate how the conditioning strategies affect the model's performance. Figure 3 presents the comparison results. As we can observe, inducing inductive bias to the conditioning strategy consistently leads to lower prediction errors compared to other conditioning approaches across various datasets. The second-best performing strategy is the "Adaptive Sum of Prompts", which employs an attention mechanism to flexibly aggregate prompts related to different aspects for each token. This approach exhibits promise in addressing diverse spatio-temporal models in a general and flexible manner. Conversely, the conditioning strategy without self-attention operation demonstrates the poorest performance. This outcome indicates the importance of incorporating the prompt within the Transformer's self-attention layers for the generative pre-training process.

**Prompts.** We conduct experiments to investigate the influence of prompt selection on the final performance. Figure 4 illustrates the comparison results of three prompt methods: the spatio-temporal prompt (ours), the spatial prompt (w/o temporal), and the temporal prompt (w/o spatial). As we can observe, eliminating either the spatial or temporal prompt leads to a reduction in the model's performance, while employing both the spatio-temporal prompt yields the most favorable performance. These findings underscore the significance of harnessing the characteristics of the target city from both spatial and temporal perspectives in the context of spatio-temporal prediction.

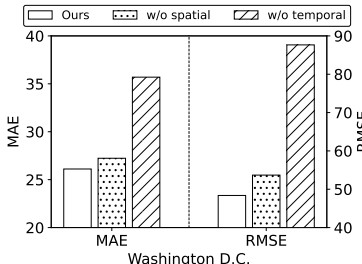

Figure 4: Performance comparison of different prompts with Washington D.C. as the target city.

## 5 CONCLUSION

GPD introduces a pioneering generative pre-training framework for spatio-temporal few-shot learning, addressing the challenges posed by data scarcity and heterogeneity in smart city applications. By conducting pre-training in the parameter space, GPD addresses the inherent disparities present in the data space across different cities and enables a new but effective knowledge transfer. Its model-agnostic nature ensures compatibility with existing urban computing models, making it a valuable tool for researchers and practitioners in the field. Our framework represents a significant advancement in urban transfer learning, which has the potential to revolutionize smart city applications in data-scarce environments and contribute to more sustainable and efficient urban development. As for future work, researchers can explore more sophisticated methods for prompt selection, such as leveraging large language models to capture the unique characteristics of cities.

ACKNOWLEDGMENTS

This work was supported in part by the National Natural Science Foundation of China under U23B2030, U22B2057 and U20B2060, and the National Key Research and Development Program of China under grant 2020YFA0711403. This work is also supported by a grant from the Guoqiang Institute, Tsinghua University under 2021GQG1005.

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

**(a) Data-space Knowledge**    **(b) Parameter-space Knowledge**

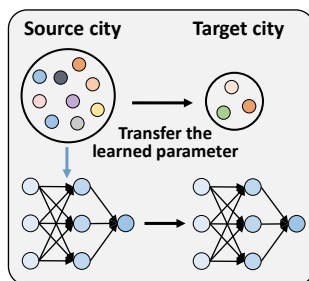 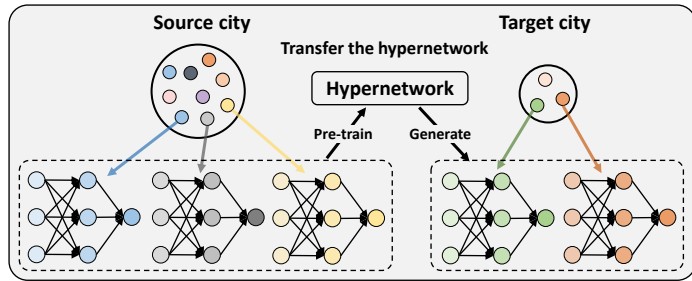

Figure 5: Comparison of data-space and parameter-space knowledge across cities. Colorful points represent divided regions in the city.

## A  METHODOLOGY DETAILS

### A.1  DIFFERENT KNOWLEDGE FOR TRANSFER

Figure 5 compares the difference between our framework and other urban transfer learning methods, which focus on parameter-space knowledge and data-space knowledge, respectively.

### A.2  REGION PROMPTS

We design region prompts to leverage auxiliary data that captures city characteristics. This prompting technique facilitates to utilize external information more flexibly. In our experiments, we adopt a granular approach by crafting distinct prompts for each region within the city. These prompts are meticulously designed to encapsulate the unique characteristics of each region. To facilitate accurate spatio-temporal predictions, we create two specific types of prompts: a spatial prompt and a temporal prompt. The spatial prompt is tailored to provide an in-depth representation of the spatial attributes, encompassing geographical features, environmental conditions, and interconnections with neighboring regions. Complementing the spatial prompt, we also introduce a temporal prompt. This prompt captures the temporal dynamics of each region.

**Spatial Prompt.**   The spatial prompt is obtained by pre-training an urban knowledge graph (UKG) (Zhou et al., 2023b), which is meticulously designed to encapsulate the extensive environmental information within a city. Specifically, we represent urban regions as distinct entities within the UKG framework. We use relations "BorderBy" and "NearBy" to capture the spatial adjacency among regions. By leveraging this adjacency representation, we aim to capture the influence that proximate regions exert on one another. Furthermore, our UKG incorporates an understanding of the functional similarity between these urban regions. This insight is quantified by computing the cosine similarity of the distribution of Points of Interest (POI) categories between region pairs. We establish a "SimilarFunc" relation to establish connections between regions that exhibit functional similarity, which emphasizes the critical role played by shared functions in shaping the urban landscape.

| Relation | Head & Tail Entity Types | Semantic Information |
|---|---|---|
| BorderBy | (Region, Region) | Regions share part of the boundary |
| NearBy | (Region, Region) | Regions are within a certain distance |
| SimilarFunc | (Region, Region) | Regions have similar functions |

Table 3: The details of relations in the urban knowledge graph.

To extract the spatial prompts from the constructed Urban Knowledge Graph (UKG), we employ a state-of-the-art KG embedding model, TuckER (Balažević et al., 2019), to learn an embedding representation for each region. TuckER evaluates the plausibility of triplets as follows:

$$\phi(h, r, t) = \mathcal{W} \times_1 e_h \times_2 e_r \times_3 e_t, \tag{3}$$

where $\mathcal{W} \in \mathbb{R}^{d_{KG}^3}$ is a learnable tensor, $\times_n$ denotes the tensor product along the $n_{th}$ dimension, and $e_h, e_r, e_t \in \mathbb{R}_{KG}^d$ are the embeddings of head entity $h$, tail entity $t$ and relation $r$ respectively. The primary objective of the KG embedding model is to maximize the scoring function for triplets that exist in the UKG, thereby preserving the knowledge contained within the UKG.

In summary, our UKG leverages the 'BorderBy' and 'NearBy' relationships to articulate spatial connections and the 'SimilarFunc' relationship to underscore functional parallels between urban regions. The benefits of UKG are twofold. First, it integrates various relationships within the city, allowing the learned embeddings of regions to provide descriptive information about their respective urban environments. Secondly, in contrast to time series data collected by sensors or GPS devices, the features utilized in the Urban Knowledge Graph (UKG) are readily available in all urban areas. This accessibility makes the UKG scalable and adaptable to cities, even those with limited development levels. The details of relations in UKG are shown in Table 3.

**Temporal Prompt.** The temporal prompt is derived through a strategic application of an unsupervised pre-training model designed for time series, as introduced by Shao et al (Shao et al., 2022b). This approach shares similarities with the concept of a Masked AutoEncoder (MAE) (He et al., 2022) for sequence data. Initially, the time series data for each region is subjected to a masking procedure, where random patches within the time series are concealed. Subsequently, an encoder-decoder model is trained on this modified data to reconstruct the original time series. This training process is centered around the objective of reconstructing the complete time series based solely on the partially observable series patches. Given that time series data often exhibits lower information density, a relatively high masking ratio (75%) is employed. This higher masking ratio is crucial for creating a self-supervised learning challenge that encourages the model to capture meaningful temporal patterns from incomplete observations. Upon successful completion of the self-supervised learning phase for time series data, the output of the encoder yields the temporal embeddings. In essence, this method capitalizes on self-supervised learning to extract valuable temporal features from time series data, which can subsequently be used as temporal prompts.

## A.3 CONDITIONING STRATEGIES

**Pre-conditioning.** "Pre" denotes that the prompt is integrated into the token sequence before being fed into self-attention layers. In this method, we simply add the region prompt $p$ to the token embeddings within the input sequence.

**Pre-conditioning with inductive bias.** In this variant, we adopt a different approach to add the region prompt $p$ to the token embeddings within the input sequence. The spatial prompt is incorporated into spatial-related parameters uniformly and the temporal prompt into temporal-related parameters uniformly as follows:

$$
\begin{aligned}
[\boldsymbol{x}_{s,1}, \boldsymbol{x}_{s,1}, \cdots, \boldsymbol{x}_{s,m}] &= [\boldsymbol{x}_{s,1}, \boldsymbol{x}_{s,1}, \cdots, \boldsymbol{x}_{s,m}] + \underbrace{[\boldsymbol{p}_s, \boldsymbol{p}_s, \cdots, \boldsymbol{p}_s]}_{m} \\
[\boldsymbol{x}_{t,1}, \boldsymbol{x}_{t,1}, \cdots, \boldsymbol{x}_{t,n}] &= [\boldsymbol{x}_{t,1}, \boldsymbol{x}_{t,1}, \cdots, \boldsymbol{x}_{t,n}] + \underbrace{[\boldsymbol{p}_t, \boldsymbol{p}_t, \cdots, \boldsymbol{p}_t]}_{n},
\end{aligned}
\tag{4}
$$

where $m$ and $n$ represent the number of tokens for spatial parameters and temporal parameters, $p_s$ denotes the spatial prompt, and $p_t$ denotes the temporal prompt.

**Pre-adaptive conditioning.** In this variant, we introduce an attention mechanism, which determines to what extent the prompt should be added to specific token embeddings. We denote the prompt as $p \in \mathbb{R}^{2 \times E}$, where $E$ is the embedding size of spatial and temporal prompts. This approach aims to empower the model to learn how to adaptively utilize the prompts, enhancing its conditioning capabilities. The utilization of the prompt can be formulated as follows:

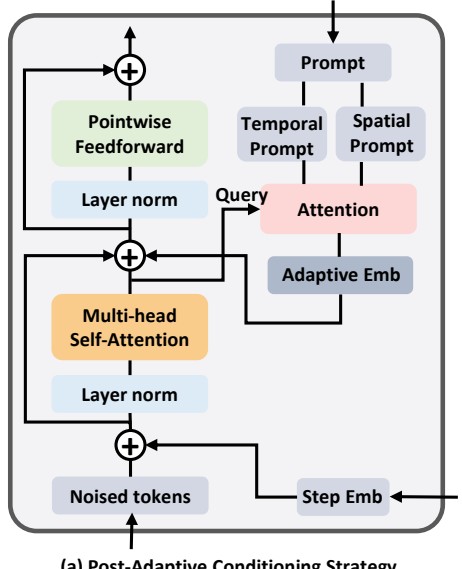 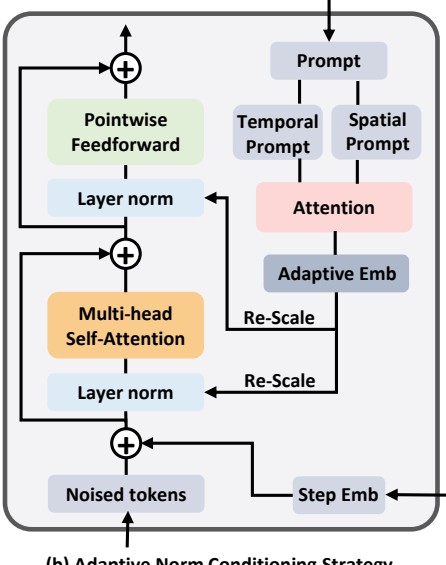

Figure 6: Illustration of two conditioning strategies: (a) "Post-Adaptive Conditioning" and (b) "Adaptive Norm Conditioning".

$$u_j = \tanh(W_w p_j + b_w), j \in \{0, 1\} \tag{5}$$

$$\alpha_{i,j} = \frac{\exp(u_j^T, x_i)}{\sum_k \exp(u_k^T, x_i)} \tag{6}$$

$$P_i = \sum_j \alpha_{i,j} p_j, j \in \{0, 1\} \tag{7}$$

where $p_0$ and $p_1$ represent the spatial prompt and temporal prompt, respectively, $P_i$ denotes the aggregated prompt from two aspects for the $i_{th}$ token.

**Post-adaptive Conditioning.** Figure 6 (a) illustrates this conditioning strategy. The aggregated prompt based on the attention mechanism is added after the multi-head self-attention in each transformer layer. Specifically, the query used for spatio-temporal attentive aggregation is the output of the multi-head self-attention layer.

**Adaptive norm conditioning.** Figure 6 (b) illustrates this conditioning strategy. The aggregated prompt based on the attention mechanism is used for re-scaling the output in each layer norm.

## A.4 NETWORK LAYERS OF SPATIO-TEMPORAL PREDICTION MODELS

We provide details of parameter tokenizers introduced in Section 3.5. In our experiments, we implement our framework on three spatio-temporal prediction models, STGCN (Yu et al., 2017), GWN (Wu et al., 2019), and STID (Shao et al., 2022a). We present how to transform their network layers into a vector-based token sequence. According to Table 4 and Table 5, the transformation from parameter layers to a token sequence can be formulated as follows:

$$
\begin{aligned}
gcd &= GCD(numel(s_1), numel(s_2), \ldots, numel(s_m)) \\
L_i &= c_i * numel(s_i)/gcd \\
L &= \sum_i L_i,
\end{aligned}
\tag{8}
$$

where $GCD$ denotes the calculation of the Greatest Common Divisor, $m$ denotes the number of different layer types, $s_i$ denotes the shape, $numel$ denotes the total number of elements in the tensor

| | Layer name | Parameter shape | Count |
|---|---|---|---|
| | Block1-TimeBlock1-conv | [32,2,1,4] | 3 |
| Block1 | Block1-GraphConv_Theta1 | [32,8] | 1 |
| | Block1-TimeBlock2-conv | [32,8,1,4] | 3 |
| TimeBlock | last_TimeBlock-conv | [32,32,1,4] | 3 |
| Linear | Fully Connected Layer | [6,96] | 1 |

Table 4: Parameter structure of a STGCN Model

| Layer name | Parameter shape | Count |
|---|---|---|
| filter_convs.x | $[32, 32, 1, 2]$ | 8 |
| gate_convs.x | $[32, 32, 1, 2]$ | 8 |
| residual_convs.x | $[32, 32, 1, 1]$ | 8 |
| skip_convs.x | $[32, 32, 1, 1]$ | 8 |
| bn.x | $[32] + [32]$ | 8 |
| gconv.x.mlp.mlp | $[32, 160, 1, 1]$ | 8 |
| start_conv | $[32, 2, 1, 1]$ | 1 |
| end_conv1 | $[32, 32, 1, 1]$ | 1 |
| end_conv2 | $[6, 32, 1, 1]$ | 1 |

Table 5: Parameter structure of a GWN Model

| | Layer name | Parameter shape | Count |
|---|---|---|---|
| Time_series_emb | conv | [32,24,1,1]+[32] | 1 |
| | Layer1-fc | [32,32,1,1]+[32] | 2 |
| EncoderLayers | Layer2-fc | [32,32,1,1]+[32] | 2 |
| | Layer3-fc | [32,32,1,1]+[32] | 2 |
| Regression_layer | conv | [6,32,1,1] | 1 |

Table 6: Parameter structure of a STID Model

$c_i$ denotes the count of this type of layer. In this way, we obtain a token sequence with length as $L$ and embedding size as $gcd$.

**STGCN (Yu et al., 2017).** Spatio-temporal graph convolution network. Different from regular convolutional and recurrent units, this model build convolutional structures on graphs. We use a 3-layer STGCN block, and utilize a 1-layer MLP as the output predictor. Table 4 shows the detailed network layers of a STGCN.

**GWN (Wu et al., 2019).** Graph WaveNet. This model developed a novel adaptive dependency matrix and learned it through node embeddings to capture the spatial dependency. It also combines with a stacked dilated causal convolution component. We use a 2-layer 4-block GWN model. Table 4 shows the detailed network layers of a GWN.

**STID (Shao et al., 2022a).** Spatio-Temporal Identity. This model is a simple Multi-Layer Perceptrons (MLPs)-based approach. By identifying the indistinguishability of samples in both spatial and temporal dimensions, it is simple yet effective for spatio-temporal prediction.

### A.5 ALGORITHMS

We present the training algorithm for spatio-temporal graph prediction in Algorithm 1. Besides, we present the training algorithm and sampling algorithm for the diffusion model in Algorithm 2 and Algorithm 3, respectively.

## B EXPERIMENT DETAILS

### B.1 DATASETS

In this section, we introduce the details of the used real-world datasets.

---

**Algorithm 1** Model Parameter Preparation

---

1: **Input**: Dataset $D = \{D_1, D_2, \ldots, D_{M_s}\}$, neural networks $F = \{f_{\theta_1}, f_{\theta_2}, \ldots, f_{\theta_{M_s}}\}$ of the spatio-temporal prediction model, loss function $L$, parameter storage $S$.
2: **Output**: Parameter storage $S$.
3: **Initialize**: Learnable parameters $\theta_m$ for $f_m$, parameter storage $S = \{\}$.
4: **for** $m \in \{1, 2, \ldots, M_s\}$ **do**
5:     **for** $epoch \in \{1, 2, \ldots, N_{iter}\}$ **do**
6:         Sample a mini-batch of inputs and labels from the dataset $D_m$ $\{x, y\} \sim D_m$
7:         Compute the predictions $\hat{y} \leftarrow f_{\theta_m}(x)$
8:         Compute the loss $\mathcal{L} \leftarrow L(\hat{y}, y)$
9:         Update the model's parameters $\theta_m \leftarrow update(\mathcal{L}; \theta_m)$
10:     **end for**
11:     Save the optimized model $S \leftarrow S \cup \{\theta_m\}$
12: **end for**

---

**Algorithm 2** Generative Pre-training of the Diffusion Model

---

1: **Input**: Parameter storage $S$, diffusion model $G$.
2: **Initialize**: Learnable parameters $\gamma$ for $G$
3: **for** $epoch \in \{1, 2, \ldots, N_{iter}\}$ **do**
4:     Sample a parameter sample $\theta_0 \sim q(\theta_0)$ and $\epsilon \sim \mathcal{N}(0, I)$
5:     Sample diffusion step $k \sim \text{Uniform}(1, \ldots, K)$
6:     Take gradient descent step on

$$\nabla_\gamma \|\epsilon - \epsilon_\gamma(\sqrt{\overline{\alpha}_k}\theta_0 + \sqrt{1 - \overline{\alpha}_k}\epsilon, p, k)\|^2$$

7: **end for**

---

**Algorithm 3** Parameter Sampling

---

1: **Input**: Gaussian noise $\theta_K \sim \mathcal{N}(0, I)$, prompts $P = \{p_1, p_2, \ldots, M\}$
2: **Output**: Parameters $\theta_t = \{\theta_{1,0}, \theta_{2,0}, \ldots, \theta_{M_t,0}\}$.
3: **Initialize**: Learnable parameters $\gamma$ for $G$, $\theta_s = \{\}$.
4: **for** $m \in \{1, 2, \ldots, M_t\}$ **do**
5:     **for** $k = K \rightarrow 1$ **do**
6:         **if** $K > 1$ **then**
7:             $z \sim \mathcal{N}(0, I)$
8:         **else**
9:             $z = 0$
10:         **end if**
11:

$$\theta_{m,k-1} = \frac{1}{\sqrt{\alpha_k}}\left(\theta_{m,k} - \frac{\beta_k}{\sqrt{1 - \overline{\alpha}_k}}\epsilon_\gamma(\theta_{m,k}, p, k)\right) + \sqrt{\beta_k}z$$

12:     **end for**
13:     $\theta_s = \theta_s \cup \theta_{m,0}$
14: **end for**

---

| Datasets | New York City | Washington, D.C. | Baltimore |
|---|---|---|---|
| #Nodes | 195 | 194 | 267 |
| #Edges | 555 | 504 | 644 |
| Interval | 1 hour | 1 hour | 1 hour |
| Time span | 2016.01.01-2016.06.30 | 2019.01.01-2019.05.31 | 2019.01.01-2019.05.31 |
| Mean | 70.066 | 30.871 | 18.763 |
| Std | 71.852 | 58.953 | 28.727 |

Table 7: The basic information and statistics of four real-world datasets for crowd flow prediction.

| Datasets | METR-LA | PEMS-BAY | Didi-Chengdu | Didi-Shenzhen |
|---|---|---|---|---|
| #Nodes | 207 | 325 | 524 | 627 |
| #Edges | 1722 | 2694 | 1120 | 4845 |
| Interval | 5 min | 5 min | 10 min | 10 min |
| Time span | 2012.05.01-2012.06.30 | 2017.01.1-2017.06.30 | 2018.1.1-2018.4.30 | 2018.1.1-2018.4.30 |
| Mean | 58.274 | 61.776 | 29.023 | 31.001 |
| Std | 13.128 | 9.285 | 9.662 | 10.969 |

Table 8: The basic information and statistics of four real-world datasets for traffic speed prediction.

### B.1.1 CROWD FLOW PREDICTION.

- **NYC Dataset.** In this dataset, we define regions as census tracts and aggregate taxi trips from NYC Open Data to derive hourly inflow and outflow information. The division of train and test regions is based on community districts. Specifically, the Manhattan borough comprises 12 community districts, and we designate 9 of them as train regions, reserving the remaining 3 for testing. Region-specific features encompass the count of Points of Interest (POIs), area, and population.

- **Washington, D.C. Dataset.** Regions in this dataset are defined as census tracts, and inflow data is calculated based on Point of Interest (POI)-level hourly visits. The partitioning of train and test regions is done by counties. Specifically, regions within the District of Columbia are selected as train regions, and regions in Arlington County are designated as test regions. This dataset comprises rich region features, including demographics and socioeconomic indicators.

- **Baltimore Dataset.** Similar to the D.C. dataset, regions, and inflow data in the Baltimore dataset are obtained in the same manner. Train regions consist of regions in Baltimore City, while test regions encompass Baltimore County. This dataset includes the same set of features as the D.C. dataset.

### B.1.2 TRAFFIC SPEED PREDICTION.

We conducted our performance evaluation using four real-world traffic speed datasets, following the data preprocessing procedures established in prior literature (Li et al., 2018; Lu et al., 2022). To construct the spatio-temporal graph, we treated each traffic sensor or road segment as an individual vertex within the graph. We then computed pairwise road network distances between these sensors. Finally, we construct the adjacency matrix of the nodes using road network distances and a thresholded Gaussian kernel.

- **METR-LA (Li et al., 2018; Lu et al., 2022).** The traffic data in our study were obtained from observation sensors situated along the highways of Los Angeles County. We utilized a total of 207 sensors, and the dataset covered a span of four months, ranging from March 1, 2012, to June 30, 2012. To facilitate our analysis, the sensor readings were aggregated into 5-minute intervals.

- **PEMS-BAY (Li et al., 2018; Lu et al., 2022).** The PEMS-BAY dataset comprises traffic data collected over a period of six months, from January 1st, 2017, to June 30th, 2017, within the Bay Area. The dataset is composed of records from 325 traffic sensors strategically positioned throughout the region.

- **Didi-Chengdu (Lu et al., 2022).** We utilized the Traffic Index dataset for Chengdu, China, which was generously provided by the Didi Chuxing GAIA Initiative. Our dataset selection encompassed the period from January to April 2018 and covered 524 roads situated within the central urban area of Chengdu. The data was collected at 10-minute intervals to facilitate our analysis.

- **Didi-Shenzhen (Lu et al., 2022).** We utilized the Traffic Index dataset for Shenzhen, China, which was generously provided by the Didi Chuxing GAIA Initiative. Our dataset selection included data from January to April 2018 and encompassed 627 roads located in the downtown area of Shenzhen. The data collection was conducted at 10-minute intervals to facilitate our analysis.

## B.2 BASELINES

- **HA.** Historical average approach models time series as a seasonal process and leverages the average of previous seasons for predictions. In this method, we utilize a limited set of target city data to compute the daily average value for each node. This historical average then serves as the baseline for predicting future values.

- **ARIMA.** Auto-regressive Integrated Moving Average model is a widely recognized method for comprehending and forecasting future values within a time series.

- **RegionTrans (Wang et al., 2019).** RegionTrans assesses the similarity between source and target nodes, employing it as a means to regulate the fine-tuning of the target. We use STGCN and GWN as its base model.

- **DASTNet (Tang et al., 2022).** Domain Adversarial Spatial-Temporal Network, which undergoes pre-training on data from multiple source networks and then proceeds to fine-tune using the data specific to the target network's traffic.

- **AdaRNN (Du et al., 2021).** This cutting-edge transfer learning framework is designed for non-stationary time series data. The primary objective of this model is to mitigate the distribution disparity within time series data, enabling the training of an adaptive model based on recurrent neural networks (RNNs).

- **MAML (Finn et al., 2017).** Model-Agnostic Meta Learning is an advanced meta-learning technique designed to train a model's parameters in a way that a minimal number of gradient updates result in rapid learning on a novel task. MAML achieves this by acquiring an improved initialization model through the utilization of multiple tasks to guide the learning process of the target task.

- **TPB (Liu et al., 2023).** Traffic Pattern Bank-based approach. TPB employs a pre-trained traffic patch encoder to transform raw traffic data from cities with rich data into a high-dimensional space. In this space, a traffic pattern bank is established through clustering. Subsequently, the traffic data originating from cities with limited data availability can access and interact with the traffic pattern bank to establish explicit relationships between them.

- **ST-GFSL (Lu et al., 2022).** ST-GFSL generates node-specific parameters based on node-level meta-knowledge drawn from the graph-based traffic data. This approach ensures that parameters are tailored to individual nodes, promoting parameter similarity among nodes that exhibit similarity in the traffic data.

## B.3 IMPLEMENTATION DETAILS

In the experiments, we set the number of diffusion steps N=500. The learning rate is set to 8e-5 and the number of training epochs ranges from 3000 to 12000. The dimensions of KG embedding and time embedding are both 128. Regarding the spatio-temporal prediction, we use 12 historical time steps to predict 6 future time steps. Our framework can be effectively trained within 3 hours and all experiments were completed on one NVIDIA GeForce RTX 4090.

## C ADDITIONAL RESULTS

### C.1 FLEXIBILITY OF GPD

To demonstrate the flexibility of our framework, we compare the performance with GWN as the base model. Figure 8 to Figure 10 illustrate the comparison results. As we can observe, our frame-

work still achieves the best performance on all datasets. The advantage of our framework is more significant for long-step predictions.

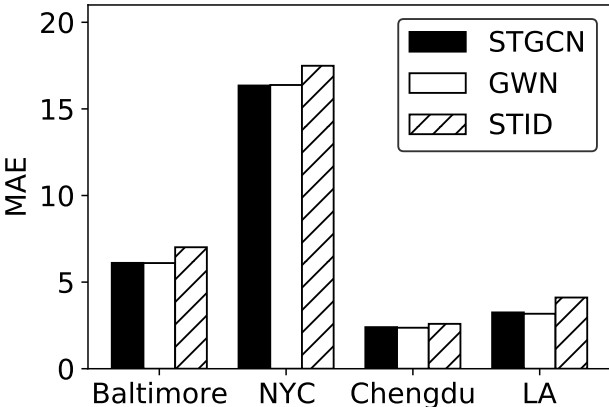

Figure 7: Performance comparison of different spatio-temporal prediction models on crowd flow prediction task (Baltimore and NYC) and traffic speed prediction task (Chengdu and LA).

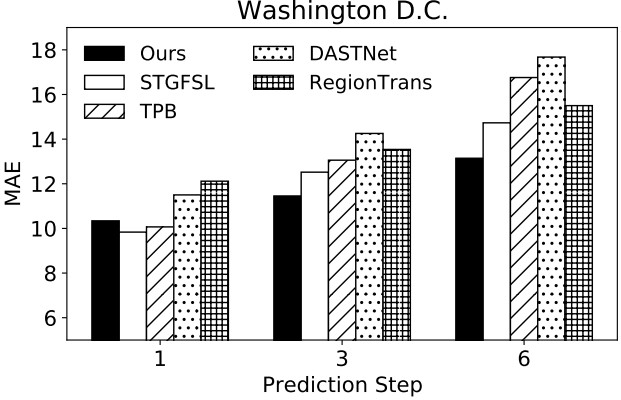

Figure 8: Performance comparison with GWN as the base model on Washington D.C. dataset.

## C.2   TIME CONSUMPTION ANALYSIS

| Model | RegionTrans | DASTNet | MAML | TPB | STGFSL | GPD |
|---|---|---|---|---|---|---|
| Computational cost | ~1.5min | ~1h | ~2h | ~1h | ~2h | ~3h |

Table 9: Training time consumption of our framework and baseline solutions.

Table 9 present a detailed comparison of the computational cost of our proposed model against baselines. Our model demonstrates efficient training, completing within 3 hours on a single GPU (RTX-2080Ti), which we believe is an acceptable time consumption. It is important to note that the slightly longer time consumption in our model is attributed to the step-by-step denoising process implemented in DDPM approach (Ho et al., 2020).

We choose DDPM because it is a classic, simple but effective diffusion framework. Notably, our framework is designed to be compatible with more efficient diffusion models, such as DDIM (Song et al., 2020). Employing these models has the potential to significantly reduce the current computational cost. We want to underscore that our framework has the ability to achieve remarkably better performance with a relatively lower overhead.

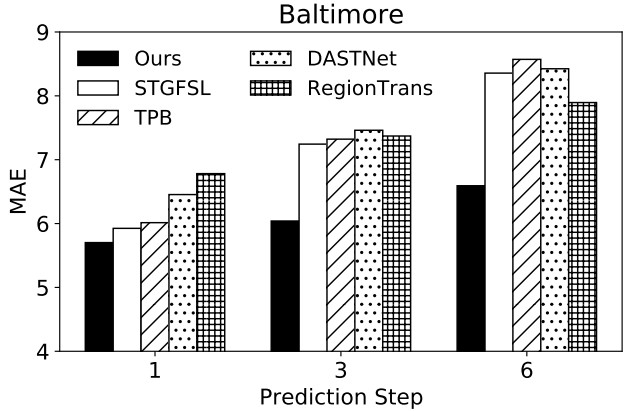

Figure 9: Performance comparison with GWN as the base model on Baltimore dataset.

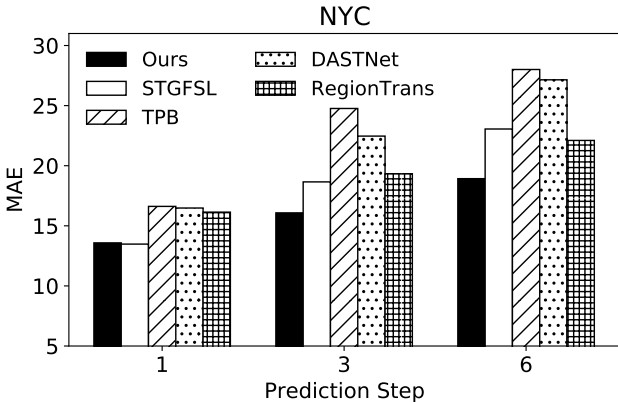

Figure 10: Performance comparison with GWN as the base model on NYC dataset.

### C.3 FINE-GRAINED PERFORMANCE ANALYSIS

We conducted fine-grained evaluations to discern the types of regions in target cities that benefit more from knowledge transfer in our pretraining framework. For this analysis, we selected Baltimore as the target city, while Washington D.C. and NYC served as source cities. To initiate our investigation, we analyzed the performance differences across various regions within Baltimore. The distribution of performance across these regions is visualized in Figure 11. This observation of varied performance across different regions prompted us to explore the factors contributing to this performance variance, aiming to gain insights into more effective knowledge transfer strategies.

In our analysis, we employed clustering on the time series data from the source cities. Figure 12 provides a visual representation of the time series data through Principal Component Analysis (PCA). Simultaneously, we conducted an in-depth analysis focusing on regions within the target city. Specifically, we selected two regions with relatively superior performance (MAE = 2.53 and 2.33) and two regions with comparatively lower performance (MAE = 30.2, 31.8). As illustrated in Figure 12, regions with better performance (depicted in red as the "good case") seamlessly align with one of the clusters identified in the source cities. In contrast, regions with lower performance (depicted in blue as the "bad case") are situated farther from the clustering centers, struggling to align with any specific clusters. This insightful observation aligns with the intuitive notion that regions sharing similar patterns are more receptive to knowledge transfer within our pretraining framework. Consequently, leveraging more diverse cities as source cities for pretraining the generative framework promises to improve overall performance. This insight is consistent with the principles observed

in Large Language Models, which often exhibit enhanced performance when trained on a highly diverse dataset (Gao et al., 2020).

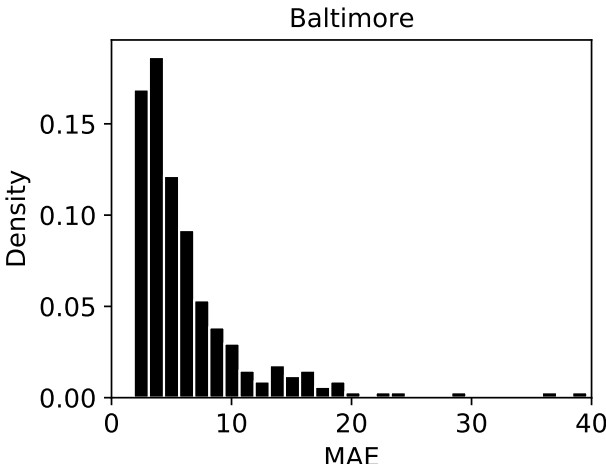

Figure 11: Performance distribution of different regions in Baltimore.

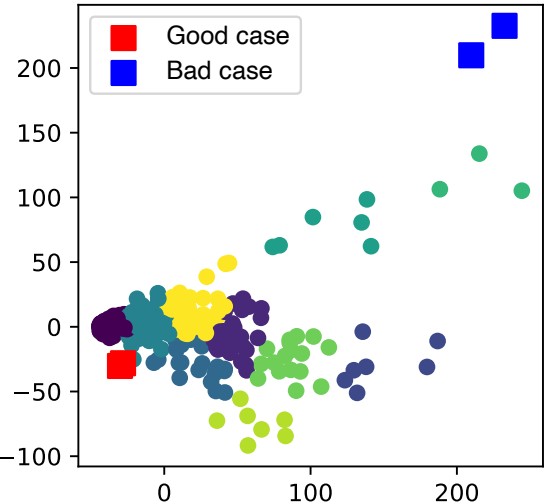

Figure 12: Clustering visualization of the source city data and the case study of the target city.

### C.4 Synthetic Data Experiments

The effectiveness of our approach is grounded in two key hypotheses:

**Assumption 1.** *Regions with distinct spatio-temporal patterns correspond to different optimized model parameters. This relationship can be mathematically expressed as follows:*

$$P_i^* \neq P_j^* \quad if \quad D(X_i, X_j) > \epsilon, \tag{9}$$

*where $P_i^*$ represents the optimized model parameters for region $i$, $X_i$ denotes the spatio-temporal pattern of the region $i$, $D$ signifies the dissimilarity metric between the patterns of two regions.*

**Assumption 2.** *There exists a mapping relation $f_\theta$ between the conditions from each region $C_i$ and the optimized model parameters $P_i^*$.*

$$P_i = f_\theta(C_i) \quad where \quad \theta = \underset{\theta}{argmin} \sum_i Distance(P_i^*, f_\theta(X_i)) \tag{10}$$

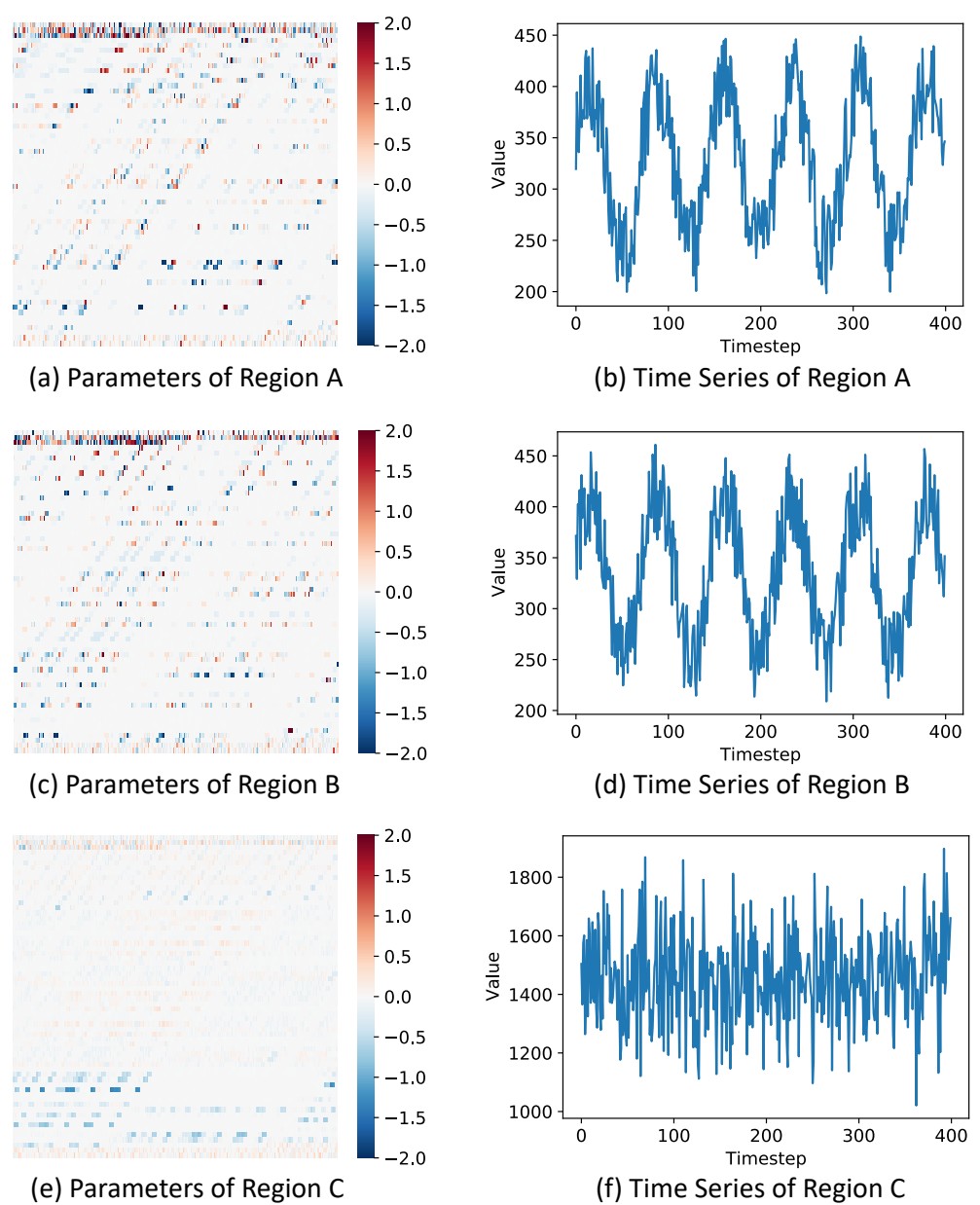

Figure 13: Visualization of parameters and time series for three regions in the source city.

*where $P_i$ is the generated model parameters.*

To empirically validate these hypotheses, we conducted experiments on two synthetic datasets, allowing us to manipulate pattern similarities between nodes. This design enables an investigation into the framework's ability to generate effective parameters. The experimental setup involves two cities: one designated as the source city, and the other as the target city, each comprising 200 regions. We first construct a graph by randomly adding edges between regions. Then, for each region, we generated a time series following specific patterns while incorporating random noise. This approach ensured that regions have distinct spatio-temporal patterns.

In Figure 13, we present the parameter maps of well-trained models for three regions in the source city. Notably, regions A (Figure 13(b)) and B (Figure 13(d)) exhibit highly similar time series patterns, whereas region C (Figure 13(f)) displays distinctly different patterns. In the meantime, the

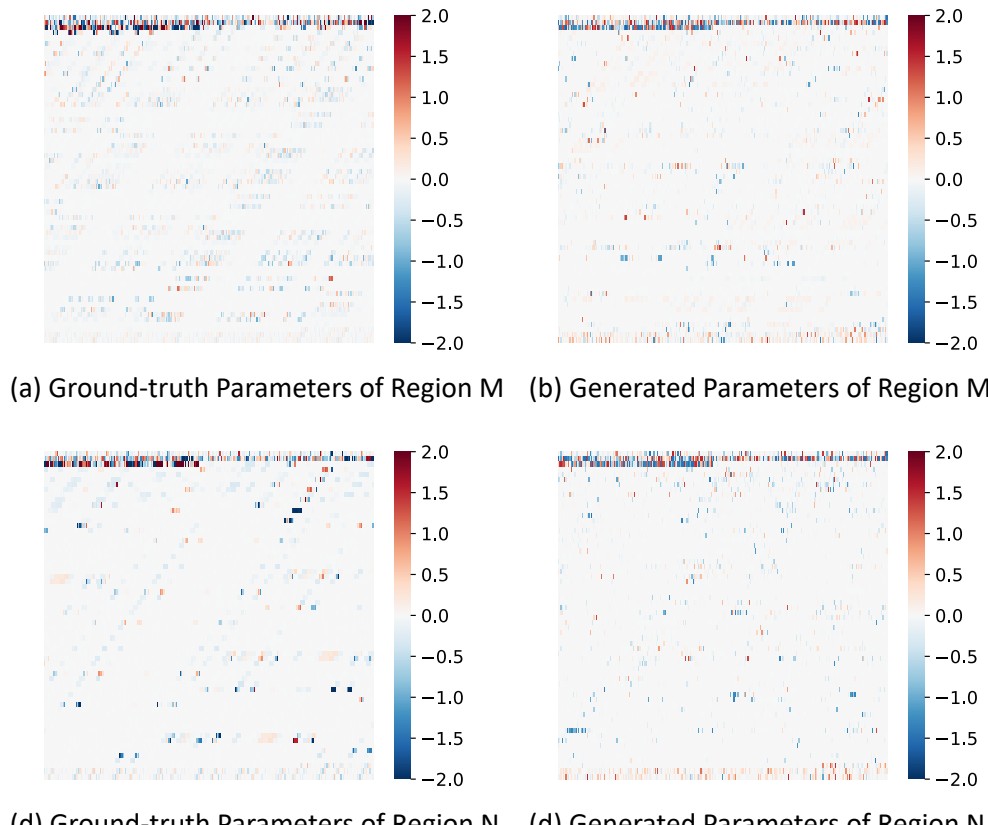

(a) Ground-truth Parameters of Region M     (b) Generated Parameters of Region M

(d) Ground-truth Parameters of Region N     (d) Generated Parameters of Region N

Figure 14: Visualization of ground-truth parameters and generated parameters for two regions in the target city.

symmetrical positions of nodes A and B in the graph (depicted in Figure 15) suggest that regions A and B have very similar spatial patterns. Therefore, we can assume that regions A and B have very similar spatio-temporal patterns. Consequently, the parameter maps for regions A and B showcase similar distributions, diverging from the parameter map of region C. This observation underscores the necessity of employing non-shared parameters for regions to effectively adapt to diverse spatio-temporal patterns.

Moving on to the parameter generation results for the target city, we selected two regions labeled as M and N, characterized by similar time series patterns and spatial connection locations (see Figure 16). Figure 14 illustrates the comparison results. Specifically, Figures 14(a) and 14(c) depict the ground truth parameter maps for these two regions, revealing similar distribution patterns. Figures 14(b) and 14(d) illustrate the generated parameter maps for the same regions. Notably, the generated parameter maps closely align with their corresponding ground-truth distributions. This comprehensive analysis of parameter generation, presented in Figures 13, Figures 14, Figure 15, and Figure 16, collectively validates the capability of our framework in effectively generating parameters that capture diverse spatio-temporal patterns across different regions.

## C.5 PREDICTION RESULTS OF THE REMAINING CITIES

Table 10 to Table 14 show the prediction results of other datasets.

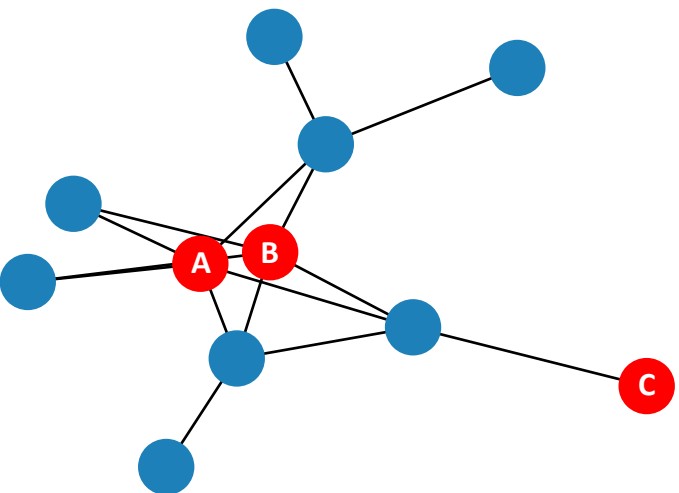

Figure 15: The sub-graph of the source city, which contains nodes A, B, and C. The graph structure denotes spatial connection relationships.

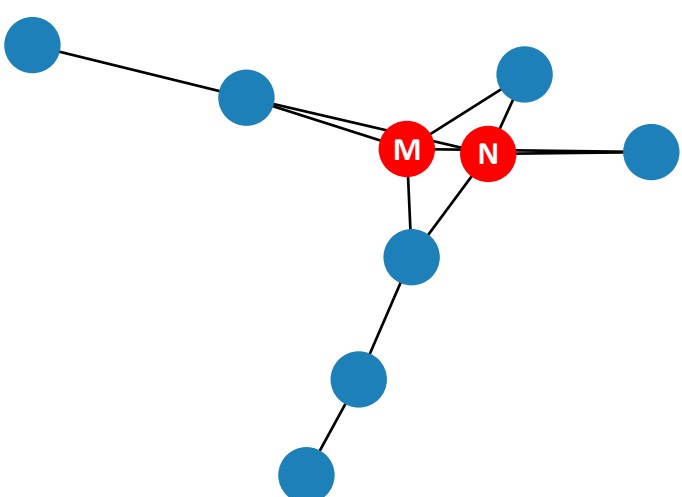

Figure 16: The sub-graph of the target, which contains nodes M and N.

| Model | MAE | | | RMSE | | |
|---|---|---|---|---|---|---|
| | Step 1 | Step 3 | Step 6 | Step 1 | Step 3 | Step 6 |
| HA | 21.520 | 21.520 | 21.520 | 47.122 | 47.122 | 47.122 |
| ARIMA | 19.703 | 15.063 | 27.083 | 37.809 | 35.093 | 61.675 |
| RegionTrans | 12.116 | 13.538 | 15.501 | 27.622 | 30.999 | 36.095 |
| DASTNet | 11.501 | 14.255 | 17.676 | 25.551 | 31.466 | 38.004 |
| MAML | 10.831 | 13.634 | 16.192 | 24.455 | 30.740 | 36.271 |
| TPB | **9.153** | _11.870_ | 15.236 | 23.420 | 28.756 | 33.424 |
| STGFSL | 9.636 | 12.178 | _14.116_ | **22.362** | _28.287_ | _33.547_ |
| GPD | _10.339_ | **11.454** | **13.144** | _23.348_ | **26.798** | **30.959** |

Table 10: Performance comparison of few-shot scenarios on Washington D.C. dataset at different prediction steps in terms of MAE and RMSE. Bold denotes the best results and underline denotes the second-best results.

| Model | MAE | | | RMSE | | |
|---|---|---|---|---|---|---|
| | Step 1 | Step 3 | Step 6 | Step 1 | Step 3 | Step 6 |
| HA | 15.082 | 15.082 | 15.082 | 26.768 | 26.768 | 26.768 |
| ARIMA | 11.150 | 12.344 | 18.557 | 19.627 | 21.665 | 35.520 |
| RegionTrans | 6.782 | 7.371 | 7.895 | 12.454 | 13.778 | _14.648_ |
| DASTNet | 6.454 | 7.461 | 8.424 | 12.304 | 13.960 | 15.225 |
| MAML | 6.765 | 8.170 | 8.834 | 13.227 | 14.470 | 16.953 |
| TPB | 6.014 | 7.322 | 8.571 | _9.832_ | 14.512 | 16.308 |
| STGFSL | _5.925_ | _7.244_ | _8.356_ | 11.157 | _13.450_ | 15.444 |
| GPD | **5.570** | **5.971** | **6.582** | **10.165** | **11.344** | **13.003** |

Table 11: Performance comparison of few-shot scenarios on Baltimore dataset at different prediction steps in terms of MAE and RMSE. Bold denotes the best results and underline denotes the second-best results.

| Model | MAE | | | RMSE | | |
|---|---|---|---|---|---|---|
| | Step 1 | Step 3 | Step 6 | Step 1 | Step 3 | Step 6 |
| HA | 34.705 | 34.705 | 34.705 | 52.461 | 52.461 | 52.461 |
| ARIMA | 27.865 | 27.695 | 33.771 | 40.643 | 45.003 | 59.359 |
| RegionTrans | 16.138 | 19.318 | _22.103_ | 26.248 | 32.489 | _37.654_ |
| DASTNet | 16.480 | 22.464 | 27.147 | 25.788 | 36.717 | 43.834 |
| MAML | 14.083 | 19.753 | 24.493 | 23.473 | 33.079 | 40.407 |
| TPB | 16.616 | 21.835 | 28.005 | **20.50** | _28.386_ | 37.701 |
| STGFSL | **13.479** | _18.654_ | 23.054 | _21.918_ | 31.106 | 37.818 |
| GPD | _13.580_ | **16.076** | **18.923** | 22.081 | **27.329** | **32.260** |

Table 12: Performance comparison of few-shot scenarios on NYC dataset at different prediction steps in terms of MAE and RMSE. Bold denotes the best results and underline denotes the second-best results.

| Model | MAE | | | RMSE | | |
|---|---|---|---|---|---|---|
| | Step 1 | Step 3 | Step 6 | Step 1 | Step 3 | Step 6 |
| HA | 3.257 | 3.257 | 3.257 | 6.547 | 6.547 | 6.547 |
| ARIMA | 7.176 | 7.262 | 7.114 | 10.84 | 11.01 | 10.89 |
| RegionTrans | 2.846 | 3.278 | 3.925 | 4.417 | 5.729 | 7.039 |
| DASTNet | 2.695 | 3.205 | 3.809 | 4.267 | 5.516 | 7.028 |
| MAML | 2.756 | 3.121 | 3.896 | 4.182 | 5.584 | 6.909 |
| TPB | **2.485** | 3.129 | 3.680 | **4.094** | 5.513 | 6.816 |
| STGFSL | 2.679 | 3.187 | 3.686 | 4.147 | 5.599 | 6.987 |
| GPD | 2.587 | **3.098** | **3.674** | 4.135 | **5.502** | **6.715** |

Table 13: Performance comparison of few-shot scenarios on METR-LA dataset at different prediction steps in terms of MAE and RMSE. Bold denotes the best results and underline denotes the second-best results.

| Model | MAE | | | RMSE | | |
|---|---|---|---|---|---|---|
| | Step 1 | Step 3 | Step 6 | Step 1 | Step 3 | Step 6 |
| HA | 3.142 | 3.142 | 3.142 | 4.535 | 4.535 | 4.535 |
| ARIMA | 5.179 | 5.456 | 5.413 | 6.829 | 7.147 | 7.144 |
| RegionTrans | 2.388 | 2.845 | 3.071 | 3.464 | 4.166 | 4.488 |
| DASTNet | 2.278 | 2.818 | 3.164 | 3.256 | 3.909 | 4.427 |
| MAML | 2.396 | 2.885 | 3.181 | 3.327 | 4.050 | 4.499 |
| TPB | 2.156 | 2.593 | 3.116 | 3.082 | 3.637 | 4.382 |
| STGFSL | 2.133 | 2.565 | 2.901 | 3.183 | 3.815 | 4.291 |
| GPD | **2.031** | **2.381** | **2.550** | **2.948** | **3.422** | **3.630** |

Table 14: Performance comparison of few-shot scenarios on Didi-Chengdu dataset at different prediction steps in terms of MAE and RMSE. Bold denotes the best results and underline denotes the second-best results.

