# OpenReview forum: "Spatio-Temporal Few-Shot Learning via Diffusive Neural Network Generation"
_ICLR.cc/2024/Conference — ICLR 2024 poster_

### Official Review · Reviewer_cSYa · 2023-10-26

**Soundness:** 3 good
**Presentation:** 3 good
**Contribution:** 3 good
**Rating:** 8
**Confidence:** 4

**Summary:**

The paper proposes a generative pre-training framework based on diffusion models, GPDiff, to generatively pre-train a set of model parameters and generate tailored model parameters using data from the source city guided by prompts. The proposed solution addresses the challenges of inter-city transfer caused by data gaps across cities.

**Strengths:**

- The authors design a novel generative pretraining framework for STG transfer problems, especially the unique angle of generating fine-grained model parameters that can serve as a good initial point for model optimizations on small-scale datasets.
- Utilizing diffusion modeling for generative pretraining is technically sound. The designed model structure based on transformers is also appropriate and flexible for supporting the model-agnostic requirement.
- The conducted experiments are sufficient to demonstrate the superiority of the proposed GPDiff framework in terms of not only prediction performance but also compatibility with different STG prediction models.
- The paper presentation and writing are good.

**Weaknesses:**

- The time consumption of this framework needs to be provided.
- Some details require further clarification. For example, how exactly is the improvement value of 9.8% calculated? As in the last part of the Introduction part, i.e., “an improvement of 9.8%”. Similar cases are in the description text of Table 2.
- I think the rationality behind the excellent performance of GPDiff is easy to understand. But I still expect some further explanations by designing some more pertinent experiments. For example, which kind of regions in target cities would be much easier to benefit from knowledge transfer in this pretraining framework?

**Questions:**

- Case Study only wrote for different regions to generate a different distribution of parameters, indicating that the parameters can be generated for different data, but can explain the parameters are effective?

- ABLATION STUDY content is too coupled, it can be split into CONTRAST STUDY, or directly changed to Study of GPDiff.

**Details Of Ethics Concerns:**

N/A.

---

> ### Author Response · Authors · 2023-11-18
> **Response to Reviewer cSYa (1/2)**
>
> Thanks for your constructive and valuable comments! We very much appreciate it. Please allow us to respond to your questions one by one as follows.
>
> **Q1: The time consumption of this framework needs to be provided.**
>
> **Response:** Thanks for this suggestion. In response, we present a detailed comparison of the computational cost of our proposed model against baselines, as the following table:
>
>
> | Model              | RegionTrans | DASTNet  | MAML   | TPB   | STGFSL | Ours   |
> | :----:             | :----:      | :----:   | :----: |:----: | :----: | :----: |
> | Computational cost |~1.5min      | ~1h      | ~2h    | ~1h   | ~2h    | ~3h    |
>
>
> Our model demonstrates efficient training, completed within 3 hours on a single GPU (RTX-2080Ti), which we believe is an acceptable time consumption. It is important to note that the slightly longer time consumption in our model is attributed to the step-by-step denoising process implemented in the DDPM approach.
>
>
> We chose DDPM because it is a classic, simple but effective diffusion framework.  Notably, our framework is designed to be compatible with more efficient diffusion models, such as DDIM [1]. Employing these models has the potential to significantly reduce the current computational cost. We want to underscore that our framework has the ability to achieve remarkably better performance with a relatively lower overhead. We have added discussions on the model's efficiency in the revised manuscript (Appendix C.2).
>
> [1] Song, Jiaming, Chenlin Meng, and Stefano Ermon. "Denoising Diffusion Implicit Models." International Conference on Learning Representations. 2020.
>
> **Q2: Some details require further clarification. For example, how exactly is the improvement value of 9.8% calculated? As in the last part of the Introduction part, i.e., “an improvement of 9.8%”. Similar cases are in the description text of Table 2.**
>
> **Response:**  The reported improvement of 9.8% is calculated by averaging the error reductions presented in Table 1 (previous version). The calculation is as follows:
>
> $$Improvment = (4.32\\% + 4.62\\% + 17.1\\% + 14.9\\% + 13.6\\% + 7.26\\% + 8.17\\% + 8.62\\%)/8=9.82\\%$$
>
>
> We have revised the presented tasks and datasets in Table 1, so the specific improvement value has also been changed to 7.87\% by averaging these four datasets. In response, we have revised the description to include information on the specific scenarios or tasks for which this improvement is observed.
>
>
> **Q3: I think the rationality behind the excellent performance of GPDiff is easy to understand. But I still expect some further explanations by designing some more pertinent experiments. For example, which kind of regions in target cities would be much easier to benefit from knowledge transfer in this pretraining framework?**
>
> **Response:** We appreciate the insightful suggestion.  In response, we have added **fine-grained evaluations**. The conclusion is that **regions sharing similar patterns are more amenable to knowledge transfer.**
>
> Upon careful analysis, we first observed performance variances among different regions in the target city. To further elucidate this phenomenon, we performed clustering on the source city time series and investigated how the target regions align with the clustering results. Our findings reveal that regions with relatively better performance seamlessly fit into one of the clusters in the source cities, whereas regions with lower performance struggle to align with any clusters. As for the detailed experiments and results, please see **Appendix C.3** in the revised manuscript.
>
> Importantly, our framework offers a valuable opportunity for highly developed cities with abundant data to assist less developed cities with limited data. We believe that leveraging more data-rich cities as source cities can lead to further performance improvements.

---

> ### Author Response · Authors · 2023-11-18
> **Response to Reviewer cSYa (2/2)**
>
> **Q4: Case Study only wrote for different regions to generate a different distribution of parameters, indicating that the parameters can be generated for different data, but can explain the parameters are effective?**
>
> **Response:** Thanks for your constructive question. In the revised version, we have designed and conducted experiments on two synthetic datasets, allowing us to control spatio-temporal pattern similarities between regions. We carefully analyzed temporal variations and spatial information. On the one hand, we observe that regions exhibiting highly similar spatio-temporal patterns showcase similar distributions of parameters.  On the other hand,  we observe that the generated parameter maps for the same regions closely align with their corresponding ground-truth distributions.
>
>
> To construct the synthetic datasets, we first construct spatial connections between regions within each city. Then, we generate time series data for these regions following specific patterns. This approach ensured that regions have distinct spatio-temporal patterns.
>
> We first explore the parameters of well-trained models for regions in the source city.  Subsequently, in the parameter generation results for the target city, we selected two regions characterized by similar time series patterns and symmetry and symmetrical spatial locations. Our analysis includes the depiction of the ground-truth parameter maps and generated parameter maps for these regions.
>
>
>
>
> For a more detailed analysis of the synthetic data experiments, please refer to **Appendix C.4** in the revised manuscript.
>
>
>
> **Q5: ABLATION STUDY content is too coupled, it can be split into CONTRAST STUDY, or directly changed to Study of GPDiff.**
>
> **Response:** Thanks for this suggestion. In the revised version, we have split the ablation study section into two subsections "4.3 Flexibility of GPDiff" and "4.4 In-depth Study of GPDiff".

---

> > ### Comment · Reviewer_cSYa · 2023-11-22
> > **Thanks for your rebuttal**
> >
> > Thanks for your detailed responses, including additional clarification and experiments.  I will maintain my score this time.

---

### Official Review · Reviewer_fmNn · 2023-10-29

**Soundness:** 2 fair
**Presentation:** 3 good
**Contribution:** 2 fair
**Rating:** 6
**Confidence:** 4

**Summary:**

This paper addresses the problem of transfer learning on spatio-temporal graphs between cities. Inspired by recent advances in NLP, the paper proposes a new pre-training approach to generate different models for various regions based on a DDPM. The major insight here is to fit different data distributions across regions and cities. The proposed framework has been evaluated on two real-world spatio-temporal applications.

**Strengths:**

1. The paper addresses an interesting real-world problem.
2. The paper is easy to follow.
3. I appreciate the design of prompts in this paper, i.e., from both spatial and temporal domains.
4. The source code has been provided.

**Weaknesses:**

Although this paper has the above merits, I would like to point out the following concerns.

W1. Insufficient related work and baselines:

The examination of related works in the paper appears to be extremely insufficient, with a noticeable lack of comparison to existing studies. From my understanding, numerous studies [1,2,3,4, 5] have delved into the application of meta-learning or hypernetworks to tailor prediction models for each node or region, especially [1] and [2] with high impacts in the field of traffic forecasting. It seems that implementing these directly could address the issue of varying data distributions across regions or cities. Therefore, I find it necessary for this paper to provide a compelling justification for the utilization of the framework proposed in this manuscript. What's more, they should be considered as baselines for empirical comparison.

W2. Limited technical novelty:

Compared to existing meta-learning-based methods, this paper has no significant novelty and technical contributions. The primary difference lies in the utilization of a DDPM (along with some prompts) to generate model parameters for each region, without theoretical contributions from the model side. This approach doesn't seem to surpass the acceptance threshold set by ICLR. The originality and technical value of this study could benefit from further enhancement.

W3. Weak evaluation:

Firstly, while the paper asserts that the proposed framework is model-agnostic, this claim is substantiated solely with evaluations on two outdated baselines, published before 2019, while recent methodologies have been overlooked, e.g., [6, 7, 8], see more in [9]. Secondly, the affirmations made in Section 4.4 lack persuasiveness. Figure 4 doesn't offer any insightful revelations, except reporting their differences. For instance, is there any proximity amongst various regions based on their respective POI information? These elements need to be elaborated on for a comprehensive understanding. Thirdly, the paper lacks a discussion on the model's efficiency. Customizing models for each region always brings considerable computational overhead. By the way, the efficiency of the proposed method should be benchmarked against the existing approaches as mentioned in W1.

W4. Some SOTA baselines are missing, such as CrossTReS, MetaST, etc.


Reference:

[1] Urban traffic prediction from spatio-temporal data using deep meta learning, KDD 2019.

[2] Adaptive graph convolutional recurrent network for traffic forecasting, NeurIPS 2020.

[3] Spatio-temporal meta learning for urban traffic prediction, TKDE 2020.

[4] Region Profile Enhanced Urban Spatio-Temporal Prediction via Adaptive Meta-Learning, CIKM 2023.

[5] Hyperst-net: Hypernetworks for spatio-temporal forecasting, arXiv.

[6] Pre-training enhanced spatial-temporal graph neural network for multivariate time series forecasting, KDD 2022.

[7] Graph Neural Controlled Differential Equations for Traffic Forecasting, AAAI 2022.

[8] Spatial-temporal identity: A simple yet effective baseline for multivariate time series forecasting, CIKM 2022

[9] Spatio-temporal graph neural networks for predictive learning in urban computing: A survey, arXiv.

**Questions:**

Please reply to W1 to W3.

Additional questions:
- Should region prompts also include information about the entire city? Merely focusing on regional features might not provide comprehensive information.
- Some alternative methods may consider transferring at different levels or scales, whereas this model appears to only consider regional features. Is there a way to adaptively incorporate large-scale features as well?
- The authors mentioned the issue of negative transfer between cities as the second limitation in the introduction. However, it is not clear how the proposed model explicitly addresses this problem. Can the authors provide more discussion and experimental evidence?

---

> ### Author Response · Authors · 2023-11-18
> **Response to Reviewer fmNn (1/5)**
>
> Thanks for your constructive and valuable comments! We very much appreciate it. Please allow us to respond to your questions one by one as follows.
>
> **Q1: Insufficient related work and baselines.**
>
> **Response:** Thanks for this comment. We appreciate your thorough examination of related work and baselines, and we would like to further clarify the distinctions between our research goals and the referenced prior work.
>
>
>
> **1. For related work**, the referenced papers [1-5] indeed share certain correlations with our framework, particularly in their use of non-shared parameters for different nodes. **It's important to clearly explain the key differences between our approach from these works.**
> - Firstly, our framework embraces the non-shared parameter concept without imposing restrictions on the spatio-temporal graph model, ensuring compatibility with state-of-the-art models. In contrast, the referenced papers propose specific model designs for spatio-temporal prediction.
> - Secondly, our approach serves as a pre-training framework, capable of acquiring knowledge from multiple cities and effectively transferring it to few-shot scenarios. In contrast, the methods you referenced lack a dedicated focus on transfer learning and have certain limitations in data-scarce urban environments, e.g., these methods necessitate training data for each city's prediction.
>
> In response to your comments, we have revisited the related work section in our initial submission. In Section 2.1, we categorized urban transfer learning works based on the granularity of transferred knowledge, distinguishing between coarse-grained and fine-grained transfer.  We acknowledge that our previous version missed a specific discussion on hypernetworks for spatio-temporal graph models [1-5]. **Your feedback prompted us to rectify this, and we have added discussions on these works in Section 2.3.**
>
>
> **2. For baselines**, It is important to note that our research addresses a different problem compared to the studies you referenced. Those papers you listed primarily focus on designing spatio-temporal graph prediction models with non-shared network parameters for individual nodes. In contrast, our work centers on the challenge of spatio-temporal graph **transfer** learning. We introduce **a general transfer learning framework** that is designed to be compatible with state-of-the-art spatio-temporal graph models.  This framework serves as a general approach rather than a specific model design.
>
> To ensure a fair comparison with other urban transfer learning methods, we implement a consistent base model for all baselines, aligning with common practices in existing transfer learning research [1]. Given the overarching nature of our framework, which is not tied to a particular model architecture, it's not appropriate to directly compare it with the specific models mentioned in your comment. While the referenced models share a similar concept (non-shared parameters) with our employed baselines like [1], our chosen baseline is more aligned with STG transfer learning, yielding state-of-the-art performance.
>
> We hope this clarification explains the motivation behind not including the mentioned papers as baselines.
>
> [1] Lu, Bin, et al. "Spatio-Temporal Graph Few-Shot Learning with Cross-City Knowledge Transfer." Proceedings of the 28th ACM SIGKDD Conference on Knowledge Discovery and Data Mining. 2022.

---

> ### Author Response · Authors · 2023-11-18
> **Response to Reviewer fmNn (2/5)**
>
> **Q2: Limited technical novelty. Compared to existing meta-learning-based methods, this paper has no significant novelty and technical contributions.**
>
> **Response:** Thanks for your comments, we would like to clarify the novelty of our paper.
>
> To address your concern about limited technical novelty, it's essential to highlight the distinct focus of our research. Unlike meta-learning-based model design, our primary objective centers around urban **transfer** learning, specifically addressing the challenge of transferability between source and target cities. Existing works in this field mainly fall into two categories: (1) feature-based transfer learning [1,2,3], which focuses on common and transferrable features, and (2) parameter-based transfer learning [4,5], utilizing meta-learning for model initialization in the target city. Comparing our work to studies within the urban transfer learning domain is more appropriate than traditional meta-learning approaches.
>
>
> The  originality and technical value of our paper are in the following aspects:
>
> **1. Fundamental and Novel Knowledge Transfer**: Unlike feature or parameter-based approaches, we propose the transfer of more fundamental knowledge—specifically, the knowledge of obtaining personalized model parameters. This represents a unique contribution to spatio-temporal graph transfer learning.
>
> **2. Pre-training Paradigm**: We successfully employ a generative pre-training paradigm for effective spatio-temporal knowledge transfer across diverse cities.
> A key departure from traditional pre-training paradigms lies in our focus on the parameter space rather than the data space.  This pre-training paradigm facilitates to acquire knowledge from data-rich cities.
>
>
> [1] Tang, Yihong, et al. "Domain adversarial spatial-temporal network: a transferable framework for short-term traffic forecasting across cities." Proceedings of the 31st ACM International Conference on Information & Knowledge Management. 2022.
> [2] Liu, Zhanyu, Guanjie Zheng, and Yanwei Yu. "Cross-city Few-Shot Traffic Forecasting via Traffic Pattern Bank." Proceedings of the 32nd ACM International Conference on Information and Knowledge Management. 2023.
> [3] Jin, Yilun, Kai Chen, and Qiang Yang. "Selective cross-city transfer learning for traffic prediction via source city region re-weighting." Proceedings of the 28th ACM SIGKDD Conference on Knowledge Discovery and Data Mining. 2022.
> [4] Finn, Chelsea, Pieter Abbeel, and Sergey Levine. "Model-agnostic meta-learning for fast adaptation of deep networks." International conference on machine learning. PMLR, 2017.
> [5] Lu, Bin, et al. "Spatio-Temporal Graph Few-Shot Learning with Cross-City Knowledge Transfer." Proceedings of the 28th ACM SIGKDD Conference on Knowledge Discovery and Data Mining. 2022.

---

> ### Author Response · Authors · 2023-11-18
> **Response to Reviewer fmNn (3/5)**
>
> **Q3: The primary difference lies in the utilization of a DDPM (along with some prompts) to generate model parameters for each region, without theoretical contributions from the model side.**
>
> **Response:** The theoretical foundation of GPDiff lies in its generative framework, where a hypernetwork is pre-trained to generate non-shared parameters for different regions in the city. The effectiveness of our approach is grounded in two key assumptions:
>
> ***Assumption 1:** Regions with distinct spatio-temporal patterns correspond to different optimized model parameters*
>
> $$P_i^* \neq P_j^* \quad \text{if} \quad D(X_i, X_j) > \epsilon,$$
>
> $$D(P_i^*, P_j^*)>D(P_m^*, P_n^*) \quad \text{if} \quad D(X_i, X_j) > D(X_m, X_n)$$
>
> where $P_i^*$ represents the optimized model parameters for region $i$, $X_i$ denotes the spatio-temporal pattern of the region $i$, and $D$ signifies the dissimilarity metric between the patterns of two regions.
>
> ***Assumption 2:** It is conceivable to find $f_\theta$ capable of capturing the mapping relation from the spatio-temporal patterns of each region ($C_i$) to the optimized model parameters ($P_i^*$). The similarity between parameters is indicative of pattern similarity.*
>
> $$P_i = f_\theta(C_i) \quad \text{where} \quad \theta = \underset{\theta}{\text{argmin}} \sum_i \text{Distance}(P_i^*, f_\theta(X_i)),$$
>
>
>
> To empirically validate these assumptions, in the revised version, we designed and conducted experiments using two synthetic datasets representing source and target cities. These datasets allow us to control spatio-temporal pattern similarities between regions.
>
>
> To confirm assumption 1, we analyzed the parameters of well-trained models for regions in the source city. Our observations revealed that regions with highly similar spatio-temporal patterns exhibit analogous parameter distributions, while regions with distinctly different patterns showcase divergent parameter distributions. This finding supports the notion that regions with distinct spatio-temporal patterns correspond to different optimized model parameters.
>
>
> To validate assumption 2, we utilized the trained generative model in the target city. We selected two regions with similar spatio-temporal patterns and observed that the generated parameter maps closely aligned with their corresponding ground-truth distributions. This indicates that our framework effectively captures the mapping relation between spatio-temporal patterns and parameter distributions.
>
>
> For a more detailed analysis of the synthetic data experiments, please refer to **Appendix C.4** in the revised manuscript.
>
>
>
> **Q4: While the paper asserts that the proposed framework is model-agnostic, this claim is substantiated solely with evaluations on two outdated baselines, published before 2019, while recent methodologies have been overlooked, e.g., [6, 7, 8], see more in [9].**
>
> **Response**: Here we would like to clarify the rationality of our experiments regarding model-agnostic characteristics.
>
> **1. As for the base model selection**, we use STGCN and GWN  because the two models have been widely recognized for their robustness and reliability.  Specifically, we adopt a consistent base model in our framework and baselines for a fair comparison, aligning with existing practices in the field [1]. STGCN and GWN were deliberately chosen to provide a strong foundation for evaluating the model-agnostic nature of our framework. Choosing them is not meant to overlook recent methodologies but rather to ensure a consistent benchmark.
>
> **2. As for the framework flexibility**, we want to emphasize that our framework extends beyond STGCN and GWN. It is adaptable to any state-of-the-art models mentioned in your review. Specifically, we implement our framework on STID [7] you referenced to demonstrate its flexibility. STID, given its relatively simpler design and potential limitations in modeling complex spatio-temporal relationships, lags slightly behind STGCN and GWN. Please see the revised section 4.3 for more details.
>
>
> In summary, the rationale behind our choice of STGCN and GWN as baselines is grounded in their well-established effectiveness, providing a robust foundation for evaluating the model-agnostic nature of our framework. In the meantime, we have also demonstrated our framework's compatibility with state-of-the-art models.
>
>
> [1] Lu, Bin, et al. "Spatio-Temporal Graph Few-Shot Learning with Cross-City Knowledge Transfer." Proceedings of the 28th ACM SIGKDD Conference on Knowledge Discovery and Data Mining. 2022.

---

> ### Author Response · Authors · 2023-11-18
> **Response to Reviewer fmNn (4/5)**
>
> **Q5: The affirmations made in Section 4.4 lack persuasiveness. Figure 4 doesn't offer any insightful revelations, except reporting their differences.**
>
>
> **Response:** Thanks for your constructive question. In the revised version, we have designed and conducted experiments on two synthetic datasets, allowing us to control spatio-temporal pattern similarities between regions. We carefully analyzed temporal variations and spatial information. On the one hand, we observe that regions exhibiting highly similar spatio-temporal patterns showcase similar distributions of parameters.  On the other hand,  we observe that the generated parameter maps for the same regions closely align with their corresponding ground-truth distributions.
>
>
> For a more detailed analysis of the synthetic data experiments, please refer to **Appendix C.4** in the revised manuscript.
>
>
> **Q6: The paper lacks a discussion on the model's efficiency. Customizing models for each region always brings considerable computational overhead. By the way, the efficiency of the proposed method should be benchmarked against the existing approaches as mentioned in W1.**
>
> **Response:** We appreciate the reviewer's comment regarding the necessity for a discussion on the model's efficiency. In response, we present a detailed comparison of the computational cost of our proposed model against baselines, as outlined in the table below:
>
>
> | Model              | RegionTrans | DASTNet  | MAML   | TPB   | STGFSL | Ours   |
> | :----:             | :----:      | :----:   | :----: |:----: | :----: | :----: |
> | Computational cost |~1.5min      | ~1h      | ~2h    | ~1h   | ~2h    | ~3h    |
>
>
> Our model demonstrates efficient training, completed within 3 hours on a single GPU (RTX-2080Ti), which we believe is an acceptable time consumption. It is important to note that the slightly longer time consumption in our model is attributed to the step-by-step denoising process implemented in the DDPM approach.
>
>
> We chose DDPM because it is a classic, simple but effective diffusion framework.  Notably, our framework is designed to be compatible with more efficient diffusion models, such as DDIM [1]. Employing these models has the potential to significantly reduce the current computational cost. We want to underscore that our framework has the ability to achieve remarkably better performance with a relatively lower overhead. We have added discussions on the model's efficiency in the revised manuscript (Appendix C.2).
>
> [1] Song, Jiaming, Chenlin Meng, and Stefano Ermon. "Denoising Diffusion Implicit Models." International Conference on Learning Representations. 2020.
>
>
> **Q7: Some SOTA baselines are missing, such as CrossTReS, MetaST, etc.**
>
> **Response:** Thanks for your observation. Regarding the absence of CrossTReS in our comparisons, it's important to note that we have no human mobility road network data for the specific cities targeted in our study. Applying CrossTReS without these crucial datasets would not result in a fair and meaningful comparison. Therefore, we did not include CrossTReS due to the incompatibility of our available data.
>
> As for MetaST, its utilization of CNN to model spatial relationships is not directly compatible with our dataset. Our dataset is structured as a graph, unlike the image-based format required by MetaST. Thus, integrating MetaST into our comparisons is not directly applicable.
>
> Despite these specific reasons for not including them as baselines in our experiments, it's crucial to clarify that we did not overlook these models. In the previous version, we did provide discussions on these state-of-the-art urban transfer learning solutions. This was strategically done in the introduction and related work sections to reveal the key contributions of our approach.
>
> **Q8: Should region prompts also include information about the entire city? Merely focusing on regional features might not provide comprehensive information.**
>
>
> **Response:** Thanks for this question. We want to clarify that our spatial prompts indeed encompass information about the entire city. These prompts are derived from an urban knowledge graph meticulously designed to encapsulate comprehensive environmental details within a city. In particular, our urban knowledge graph employs relations such as "BorderBy" and "NearBy" to capture the spatial adjacency among regions, ensuring that the information extends beyond individual regions to cover the entire city. Additionally, we consider functional similarities between urban regions. The chosen relations are selected for their high accessibility and relevance across all urban areas.
>
>
>
> For a more in-depth understanding of our spatial prompts and the incorporation of city-wide information, please refer to Appendix A.2, where we provide detailed insights into the design and implementation of our urban knowledge graph.

---

> ### Author Response · Authors · 2023-11-18
> **Response to Reviewer fmNn (5/5)**
>
> **Q9: Some alternative methods may consider transferring at different levels or scales, whereas this model appears to only consider regional features. Is there a way to adaptively incorporate large-scale features as well?**
>
> **Response:** Thanks for this question.  Our framework's prompt design is inherently flexible, allowing practitioners to adaptively incorporate various features for different scenarios.  Practitioners have the freedom to adjust the transfer level or scale as long as they have corresponding scale-based prompts from the chosen features. For example, a larger-scale urban knowledge graph can be leveraged to obtain hierarchical prompts.
>
>
> **Q10: The authors mentioned the issue of negative transfer between cities as the second limitation in the introduction. However, it is not clear how the proposed model explicitly addresses this problem. Can the authors provide more discussion and experimental evidence?**
>
> **Response:** Thanks for this question. The negative transfer between cities can be attributed into two aspects: (1) different cities exhibit varied patterns, and (2) diverse patterns exist even among different regions within a city. To mitigate negative transfer, we introduce a generative pre-training framework that tailors optimized, non-shared parameters for each region.
>
> Specifically, we addressed the negative transfer grounded in the validation of two critical assumptions as we have responded to Q3. To empirically validate these assumptions, in the revised version, we designed and conducted experiments using two synthetic datasets. It allows us to control spatio-temporal pattern similarities between nodes. The experiments, added to **Appendix C.4** in the revised manuscript, provide valuable insights into the effectiveness of our approach in handling negative transfer.
>
> Additionally, our experiments demonstrate that fine-grained transfer learning solutions, such as TPB and STGFSL, outperform coarse-grained solutions (refer to Table 1). This observation further supports the impact of negative transfer when directly transferring an overall model from one city to another.
>
>
> In the revised version, we have modified the corresponding part in the introduction.

---

> ### Author Response · Authors · 2023-11-21
> **Willing to further clarify your remaining concerns**
>
> Dear reviewer,
>
> The discussion period is almost ending. Could you please confirm whether our responses have alleviated your concerns?
>
> For your interest, we (1) clarify the difference in problem definition and methodology from your referenced papers; (2) emphasize our key contribution; (3) add theoretical analysis on synthetic data; (4) clarify the regional feature utilization. Please refer to the details in both the responses for you and the general comments.
>
> If you have further comments, we are also very happy to have a discussion. Thank you very much!

---

> > ### Comment · Reviewer_fmNn · 2023-11-22
> > **Response to authors**
> >
> > Thank you for addressing my queries. I appreciate the inclusion of more related work for discussion in the revision, and I'm pleased to see the efficiency issue has been resolved. While I still have some reservations regarding the technical novelty of the paper, I raised my score accordingly.
> >
> > I would like to suggest considering another relevant work for reference:
> > [1] GPT-ST: Generative Pre-Training of Spatio-Temporal Graph Neural Networks. Presented at NeurIPS 2023.

---

> ### Author Response · Authors · 2023-11-23
> **Thank you for suggesting the relevant work**
>
> Thank you for suggesting the consideration of the relevant work [1]. We appreciate your acknowledgment of the revisions made in our response.
>
> In comparing our work to [1], we note a shared concept of generative pre-training for spatio-temporal graph learning.  However, notable distinctions exist: **(1) Different Pretraining Targets.** While [1] employs a mask autoencoder for learning spatio-temporal dependencies, our work focuses on the generative pre-training of model parameters. **(2) Different Objecties.** Our approach proposes a general framework for urban **transfer** learning. In contrast, [1] develops a customized temporal encoder and a hierarchical spatial encoder to for spatio-temporal predictions.
>
> **We have incorporated a discussion on this relevant work in our related work section.** We hope that this enhances the clarity regarding our contributions.

---

> ### Author Response · Authors · 2023-11-23
> **Technical Novelty and Contribution**
>
> We thank the reviewer for the comment. We have clearly summarized the novelty of the method, which tackled non-trivial challenges and made contributions to STG transfer learning. Moreover, as also mentioned by other reviewers, the core novelty is "It is a pioneering practice in handling urban data-scarce scenarios that explores the paradigm of pretraining and prompt-based finetuning." "The novelty of the proposed GPDiff framework is sound." "The authors design a novel generative pretraining framework for STG transfer problems." We hope that our responses have appropriately addressed your concerns.

---

### Official Review · Reviewer_yfzH · 2023-10-31

**Soundness:** 3 good
**Presentation:** 3 good
**Contribution:** 3 good
**Rating:** 6
**Confidence:** 5

**Summary:**

This paper provides a generative pretraining framework for spatio-temporal graph (STG) transfer learning tasks in smart city applications. Unlike traditional transfer learning approaches in this field, authors leverage the power of pretraining and design a novel approach that first pretrains a diffusion-based hypernetwork and then, based on the spatio-temporal prompt, directly generates model parameters for each region in a target city. Experiments on 7 real-world datasets covering two typical STG prediction tasks, i.e., crowd flow prediction and traffic speed prediction, demonstrate the superiority of the proposed GPDiff framework over SOTAs, with an improvement of 9.8%.

**Strengths:**

S1.	This paper provides a promising solution for an important research problem. Specifically, it is a pioneering practice in handling urban data-scarce scenarios that explores the paradigm of pretraining and prompt-based finetuning.
S2.	The novelty of the proposed GPDiff framework is sound. Unlike existing works, it tackles the STG transfer learning problem from a new angle, i.e., pretraining a generative hypernetwork that captures the region-conditional distribution of optimized model parameters. This design overcomes the difficulties in applying pretraining in STG learning.
S3.	Building spatial prompts with easy-to-accessible data like POIs and region attributes is reasonable and practical. Building temporal prompts with a self-supervised representation learning technique is also technically sound.
S4.	By leveraging the generation power of transformer-based diffusion models, GPDiff achieves sota performance on two typical STG prediction tasks, covering 7 real-world datasets.

**Weaknesses:**

W1.	The presentation of experimental results can be improved. For example, in Figure 4, why only show the temporal variations? What about spatial information? Visualization of model parameters needs to be more apparent.
W2.	I find that the presented results on the traffic speed prediction task seem much less than those of the crowd flow prediction task.

**Questions:**

Please refer to the weaknesses part.

**Details Of Ethics Concerns:**

N/A.

---

> ### Author Response · Authors · 2023-11-18
> **Response to Reviewer yfzH**
>
> Thanks for your constructive and valuable comments! Please allow us to respond to your questions one by one as follows.
>
> **Q1: The presentation of experimental results can be improved. For example, in Figure 4, why only show the temporal variations? What about spatial information? Visualization of model parameters needs to be more apparent.**
>
>
> **Response:** Thanks for your constructive question. In the revised version, we have designed and conducted experiments on two synthetic datasets, allowing us to control spatio-temporal pattern similarities between regions. We carefully analyzed temporal variations and spatial information. On the one hand, we observe that regions exhibiting highly similar spatio-temporal patterns showcase similar distributions of parameters.  On the other hand,  we observe that the generated parameter maps for the same regions closely align with their corresponding ground-truth distributions.
>
>
> To construct the synthetic datasets, we first construct spatial connections between regions within each city. Then, we generate time series data for these regions following specific patterns. This approach ensured that regions have distinct spatio-temporal patterns.
>
> We first explore the parameters of well-trained models for regions in the source city.  Subsequently, in the parameter generation results for the target city, we selected two regions characterized by similar time series patterns and symmetry and symmetrical spatial locations. Our analysis includes the depiction of the ground-truth parameter maps and generated parameter maps for these regions.
>
>
> For a more detailed analysis of the synthetic data experiments, please refer to **Appendix C.4** in the revised manuscript.
>
>
> **Q2: I find that the presented results on the traffic speed prediction task seem much less than those of the crowd flow prediction task.**
>
> **Response:** Thanks for your comments. In the previous version, due to the page limit, we only presented most results on the crowd flow prediction task. We acknowledge the need for a more comprehensive representation of our model's performance on the traffic speed prediction task.
>
>
> In the revised version, we have added the experimental results on the traffic speed prediction task to the main paper and moved some traffic speed prediction results to the Appendix. The updated version now includes additional experimental results on the traffic speed prediction task, providing a more balanced result presentation. Please refer to the revised Table 1 and Figure 7, and the corresponding analysis in the main paper.

---

### Official Review · Reviewer_zXEa · 2023-11-04

**Soundness:** 3 good
**Presentation:** 3 good
**Contribution:** 3 good
**Rating:** 6
**Confidence:** 5

**Summary:**

The proposed GPDiff framework in this paper aims to tackle the challenging transfer learning problem in spatio-temporal graph (STG) predictions. At first, it basically trains a transformer-based diffusion model on collected data in source cities, to capture the mapping function from a set of region-specific features to a set of model parameters. Then GPDiff can directly generate corresponding model parameters for unseen regions in new cities, based on the spatio-temporal features of these regions. The first stage is described as the pretraining and the second stage is the prompt-based finetuning. Experiments on several datasets illustrate that GPDiff empirically performs well on different prediction tasks, compared with a series of state-of-the-art methods.

**Strengths:**

1.	Generally, the proposed method is sound and reasonable.
2.	The proposed framework seems flexible, as it can adapt to different model structures and support training on data collected from any number of source cities.
3.	The results show a large improvement. Ablation studies and case studies seem sufficient.
4.	I have checked the released code and found that it is well organized. The reproducibility should be good.

**Weaknesses:**

1. While the proposed method is sound, it is not clear what is the theoretical foundation behind GPDiff.
2. How do we guarantee that model parameters of regions among different cities are within the same parameter space?
3. Why choose diffusion for parameter generation? How much time does it consume in the denoising process?
4. The experimental setup is not clear, which confused me why only DiDi-Chengdu is used for speed experiments. Besides, the caption of table 1 is not correct if didi-cheng is for speed evaluation.
5. It seems that the method uses a newly proposed network. It is unclear whether the performance gain comes from the Diffusion or the network. Could the framework generalized to other networks?

**Questions:**

please check the weakness

---

> ### Author Response · Authors · 2023-11-18
> **Response to Reviewer zXEa (1/2)**
>
> Thanks for your constructive feedback. We have summarized your comments into four questions. We hope the following responses can address your concerns.
>
> **Q1: It is not clear what is the theoretical foundation behind GPDiff.**
>
> **Response:**    Thank you for raising this question. The theoretical foundation of GPDiff lies in its generative framework, where a hypernetwork is pre-trained to generate non-shared parameters for different regions in the city. The effectiveness of our approach is grounded in two key assumptions:
>
> ***Assumption 1:** Regions with distinct spatio-temporal patterns correspond to different optimized model parameters*
>
> $$P_i^* \neq P_j^* \quad \text{if} \quad D(X_i, X_j) > \epsilon,$$
>
> $$D(P_i^*, P_j^*)>D(P_m^*, P_n^*) \quad \text{if} \quad D(X_i, X_j) > D(X_m, X_n)$$
>
> where $P_i^*$ represents the optimized model parameters for region $i$, $X_i$ denotes the spatio-temporal pattern of the region $i$, and $D$ signifies the dissimilarity metric between the patterns of two regions.
>
> ***Assumption 2:** It is conceivable to find $f_\theta$ capable of capturing the mapping relation from the spatio-temporal patterns of each region ($C_i$) to the optimized model parameters ($P_i^*$). The similarity between parameters is indicative of pattern similarity.*
>
> $$P_i = f_\theta(C_i) \quad \text{where} \quad \theta = \underset{\theta}{\text{argmin}} \sum_i \text{Distance}(P_i^*, f_\theta(X_i)),$$
>
>
>
> To empirically validate these assumptions, in the revised version, we designed and conducted experiments using two synthetic datasets representing source and target cities. These datasets allow us to control spatio-temporal pattern similarities between regions.
>
>
> To confirm assumption 1, we analyzed the parameters of well-trained models for regions in the source city. Our observations revealed that regions with highly similar spatio-temporal patterns exhibit analogous parameter distributions, while regions with distinctly different patterns showcase divergent parameter distributions. This finding supports the notion that regions with distinct spatio-temporal patterns correspond to different optimized model parameters.
>
>
> To validate assumption 2, we utilized the trained generative model in the target city. We selected two regions with similar spatio-temporal patterns and observed that the generated parameter maps closely aligned with their corresponding ground-truth distributions. This indicates that our framework effectively captures the mapping relation between spatio-temporal patterns and parameter distributions.
>
>
> For a more detailed analysis of the synthetic data experiments, please refer to **Appendix C.4** in the revised manuscript.
>
>
>
> **Q2: How do we guarantee that model parameters of regions among different cities are within the same parameter space?**
>
> **Response:** We have some operations for regions among different cities. We initialize the model parameters from the same distribution. Data is normalized into the [-1,1] range, providing a standardized input for model training.  Different region-based model parameters serve as training data, which is a sample from the parameter space.
>
> As for the prompt design, we have certain operations to guarantee the prompt within the same parameter space. For instance, temporal prompts for regions across cities are derived from the same encoder, ensuring consistency in the prompt generation process. We construct inter-city relations to establish connections between different cities. The urban knowledge graph is trained in a unified manner, providing a cohesive understanding of relations across cities.
>
> In summary, our approach strikes a balance between ensuring parameter space consistency and allowing for the necessary flexibility to capture variations among different cities.

---

> ### Author Response · Authors · 2023-11-18
> **Response to Reviewer zXEa (2/2)**
>
> **Q3: Why choose diffusion for parameter generation? How much time does it consume in the denoising process?**
>
> **Response:**  We choose diffusion for parameter generation based on the following considerations:
>
> **1. Structural Similarity with Images.** There exists a structural similarity between parameter maps and images. Extensive research has demonstrated the superior performance of diffusion models in the context of image generation. Leveraging this similarity, we find diffusion models to be well-suited for capturing the intricate patterns and relationships within parameter maps.
>
> **2. Relevance of Diffusion Process to Parameter Optimization.** The denoising process, as employed in our framework, closely mirrors the progression from random initialization to trained parameters. This progression aligns with the optimization journey, making the diffusion process an appropriate and meaningful mechanism for generative pre-training of model parameters.
>
>
> **Regarding the time consumption during the denoising process,** our current implementation takes less than 1 minute to complete a single denoising iteration. Notably, the denoising process for different nodes can be computed in parallel, contributing to the overall efficiency of our approach. Thus, the use of the diffusion model demonstrates an acceptable time consumption. Meanwhile, it's essential to note that the required time can vary based on the choice of diffusion models.
>
> In our implementation, we use DDPM simplicity. Although DDPM serves our purposes effectively, it's recognized as relatively less efficient due to the step-by-step denoising process. If prioritizing time efficiency in denoising, alternatives like DDIM [1] could be explored.
>
> [1] Song, Jiaming, Chenlin Meng, and Stefano Ermon. "Denoising Diffusion Implicit Models." International Conference on Learning Representations. 2020.
>
>
>
> **Q4: The experimental setup is not clear, which confused me why only DiDi-Chengdu is used for speed experiments. The caption of table 1 is not correct if didi-cheng is for speed evaluation.**
>
> **Response:**  We sincerely apologize for the confusion regarding the experimental setup and the caption of Table 1, and we appreciate your attention to these details. Here we would like to provide clarification:
>
> **1. Experimental Setup Clarification.**   Our study has two prediction tasks: crowd flow prediction and traffic speed prediction. For each task, we have leveraged several datasets from different cities to ensure the robustness and generalizability of our model. Due to space constraints within the main paper, we have presented the results of three datasets for crowd flow prediction and one dataset for traffic speed prediction. The detailed results for additional datasets are included in Appendix C.5.
>
> **2. Correction to Table 1 Caption.** We are sorry for the mistake in the caption of Table 1. The correct caption should be: "Performance comparison of few-shot scenarios on two crowd flow datasets (Washington D.C. and Baltimore) and two traffic speed datasets (LA and Didi-Chengdu) in terms of MAE and RMSE."
>
>
>
> **Q5: It is unclear whether the performance gain comes from the Diffusion or the network. Could the framework generalized to other networks?**
>
> **Response:**  Given that our approach and baselines employ the same base model, the performance gain comes from the novel framework, and the core idea is a generative hypernetwork based on the diffuson model.  The baselines cannot leverage the power of generative modeling and the capability of diffusion probabilistic model.
>
>
> Our framework can be generalized to other networks. We have implemented different base models, including STGCN, GWN, and STIN (see Section 4.3 and Appendix C.1). Our framework consistently achieves superior performance, demonstrating the adaptability and effectiveness of our framework. This underscores the adaptability and effectiveness of our framework across various network architectures.

---

### Author Response · Authors · 2023-11-18
**General Comments by Authors**

We sincerely appreciate the insightful comments provided by all reviewers, and we would like to address the key novelty of our work while highlighting the distinctions from the references mentioned by Reviewer fmNn.


***1. Key Novelty.***

Our key novelty lies in the introduction of a generative pre-training framework for STG transfer learning. We have successfully implemented effective pre-training by focusing on the parameter space. This, combined with elaborated spatio-temporal prompts, enables us to achieve superior performance in the realm of urban transfer learning.

***2. Contribution to STG transfer learning.***

Our contributions to the field of STG transfer learning can be summarized in two key aspects:

- ***Fundamental and Novel Knowledge Transfer***: Unlike feature or parameter-based approaches, we propose the transfer of more fundamental knowledge—specifically, the  intricate generative process of customized model parameters. This represents an important contribution to STG transfer learning.
- ***Pre-training Paradigm***: We successfully employ the powerful generative pre-training paradigm for effective spatio-temporal knowledge transfer across diverse cities. This is achieved by shifting the pre-training target, focusing on the parameter space rather than the data space. This paradigm facilitates knowledge acquisition from data-rich cities.


***3. Differences from works referenced by Reviewer fmNn***

- ***Difference in Problem Definition.*** The works [1-5] referenced by Reviewer fmNn are primarily centered around enhancing spatio-temporal prediction tasks.  While they may incorporate non-shared parameters for individual nodes, their primary objective is to improve accuracy within a specific city. In contrast, our work is distinctly focused on urban transfer learning, specifically aiming to transfer knowledge from data-rich cities to those with limited data availability
- ***Difference in Methodology.*** [1-5] referenced by Reviewer fmNn focus on specific model designs. Although they learn node-level non-shared parameters, the learning process is intertwined with the prediction task and supervised by the training data in the city. This design limitation constrains their adaptability to cities with insufficient data. In contrast, our approach introduces a pre-training mechanism that not only aligns seamlessly with SOTA STG models but also operates independently of the spatio-temporal prediction training task.
- ***Insights from Hypernetworks.*** Our work draws unique insights from hypernetworks, inspiring us to design a pre-training mechanism from the parameter perspective rather than the data perspective.  In response to the reviewer's references, we have incorporated discussions on related works in Section 2.3 of the revised manuscript.

The revised part is highlighted in red in the manuscript.

---

### Meta-Review · Area_Chair_Xihe · 2023-12-06

**Metareview:**

The paper presents a framework called GPDiff for tackling the transfer learning problem in STG predictions. GPDiff utilizes a transformer-based diffusion model to pretrain on data from source cities and generate model parameters for regions in target cities. The framework is applied to different prediction tasks and outperforms several state-of-the-art methods, showing empirical success in STG transfer learning.

Strengths:
* The proposed framework is considered sound and reasonable, providing a promising solution to an important research problem. GPDiff is flexible and can adapt to different model structures, making it versatile for various applications and data from multiple source cities.
* The paper includes results that demonstrate a significant improvement over existing methods, supported by ablation studies and case studies.
* The paper has been criticized for its insufficient discussion of related work and baselines. The reviewers suggested comparing the proposed framework with existing studies, such as those on meta-learning or hypernetworks, which have been applied to tailor prediction models for regions. The authors provided a compelling justification for choosing GPDiff over these approaches.

Weaknesses:
* The choice of diffusion for parameter generation is not well-justified, and the time consumption during the denoising process is not addressed.
* The experimental setup is not well-documented, leading to confusion regarding why only one dataset (DiDi-Chengdu) is used for speed experiments.

**Justification For Why Not Higher Score:**

The paper introduces a promising framework for STG transfer learning but faced some criticisms regarding theoretical foundations, justification for method choice, and evaluation, most of which were addressed by the authors during the rebuttal phase.

**Justification For Why Not Lower Score:**

Unlike the referenced meta-learning-based methods that focus on STG prediction within a specific city, this work addresses a different research problem: transfer learning across cities. The new framework is compatible with many SOTA STG models.

---

### Decision · Program_Chairs · 2024-01-16

Accept (poster)